



# The contribution of melt ponds to enhanced Arctic sea-ice melt during the Last Interglacial

Rachel Diamond[1,2], Louise C. Sime[1], David Schroeder[3], and Maria-Vittoria Guarino[1,4]

[1]British Antarctic Survey, Cambridge, UK
[2]Imperial College London, London, UK
[3]Department of Meteorology, University of Reading, Reading, UK
[4]Faculty of Engineering and Physical Sciences, University of Leeds, Leeds, UK

**Correspondence:** Rachel Diamond and Louise Sime (rachel.diamond17@imperial.ac.uk and lsim@bas.ac.uk)

**Abstract.**

HadGEM3 is the first coupled climate model to simulate an ice-free Arctic during the Last Interglacial (LIG), 127 000 years ago. This simulation appears to yield accurate Arctic surface temperatures during the summer season. Here, we investigate the causes and impacts of this extreme simulated ice loss. We find that the summer ice melt is predominantly driven by thermo-
dynamic processes: atmospheric and ocean circulation changes do not significantly contribute to the ice loss. We demonstrate these thermodynamic processes are significantly impacted by melt ponds, which form on average 8 days earlier during the LIG than during the pre-industrial control (PI) simulation. This relatively small difference significantly changes the LIG surface energy balance, and strengthens the albedo feedback. Compared to the PI simulation: in mid-June, of the absorbed flux at the surface over ice-covered cells (ice concentration>0.15), ponds account for 45-50%, open water 45%, and bare ice and snow
5-10%. We show that the simulated ice loss leads to large Arctic sea surface salinity and temperature changes. The sea surface temperature and salinity signals we identify here provide a means to verify, in marine observations, if and when an ice-free Arctic occurred during the LIG. Strong LIG correlations between spring melt pond and summer ice area indicate that, as Arctic ice continues to thin in future, the spring melt pond area will likely become an increasingly reliable predictor of the September sea-ice area. Finally, we note that models with explicitly modelled melt ponds seem to simulate particularly low LIG sea ice
extent. These results show that models with explicit (as opposed to parameterised) melt ponds can simulate very different sea-ice behaviour under forcings other than the present-day. This is of concern for future projections of sea-ice loss.

## 1 Introduction

Interglacials are periods of globally higher temperatures which occur between cold glacial periods (Sime et al., 2009; Otto-Bliesner et al., 2013; Fischer et al., 2018). Glacial-interglacial cycles are largely driven by changes in the Earth's orbit which
affects incoming radiation. The Eemian period, here called the Last Interglacial or LIG, occurred 130,000-116,000 years ago. At high latitudes orbital forcing led to summertime top-of-atmosphere short-wave (TOA SW) radiation 60–75 $Wm^{-2}$ greater for the LIG, compared to the preindustrial (PI) period. This drove differences between the LIG and PI surface energy balance. Whilst the significance of these PI to LIG differences vary between climate models (Lunt et al., 2013; Otto-Bliesner et al.,





2013), proxy records of the LIG indicate that mean Arctic summer land temperature was + 4-5 °K higher than the PI (CAPE
members, 2006).

Prior to 2020, most climate models simulated LIG temperatures which were too cool compared with LIG temperature data
(Otto-Bliesner et al., 2013; IPCC, 2013). Recently, Guarino et al. (2020b) find that the loss of Arctic sea-ice in the summer
likely drove these warm Arctic temperatures, and thus suggest that previous climate models may not have simulated the loss
of enough Arctic sea-ice during the LIG. Kageyama et al. (2020) explore the ocean core based proxy records of LIG Arctic
sea-ice change. They find that sea-ice changes are more difficult to determine than temperature changes, with some conflicting
interpretations of proxy data from the available records, and imprecision in dating materials from cores in the high Arctic. This
makes it difficult to determine from these preserved biological data the mechanisms, or distribution of, sea-ice loss during the
LIG.

In terms of understanding mechanisms driving possible Arctic sea-ice change during the LIG, three main factors are known
to affect summer sea-ice behaviour: cloud cover feedbacks (Kay and Gettelman, 2009), ocean heat transport changes (Årthun
et al., 2019; Auclair and Tremblay, 2018), and albedo feedbacks (Curry et al., 1995). As well as these, changes to sea-ice
distribution caused by changes to wind patterns and ocean circulation may also affect summer sea-ice extent (Kageyama et al.,
2020; Deser and Teng, 2008; Wang et al., 2009). The Coupled Model Intercomparison Project Phase 6 (CMIP6) Palaeoclimate
Model Intercomparison Project Phase (PMIP4) or CMIP6-PMIP4 LIG experimental protocol prescribes differences between
the LIG and PI in orbital parameters, as well as differences in trace greenhouse gas concentrations (Otto-Bliesner et al., 2017).
This standardised climate modelling protocol enables the community to use models to explore these mechanisms using a
multi-model approach.

Twelve models ran the CMIP6-PMIP4 LIG simulation. All twelve models showed a substantial reduction in LIG Arctic sea-
ice compared to the PI (Kageyama et al., 2020). They yielded a minimum Arctic sea-ice area which ranged between 0.02 - 5.65
million km$^2$. Whilst inter-model differences were variously attributed to differences in the albedo feedback, ocean circulation
and heat transport, atmospheric circulation, and cloud cover, these aspects have however not yet been fully analysed for all the
models which ran the simulation (Guarino et al., 2020b; Kageyama et al., 2020; Otto-Bliesner et al., 2020).

Of these twelve models, Guarino et al. (2020b) show that the model HadGEM3 gives a good match with proxy temperatures:
the average LIG temperature anomaly in HadGEM3, for all locations with observations, is $+4.9 \pm 1.2$ K compared with the
observational mean of $+4.5 \pm 1.7$ K. This model apparently also has a good match with all, except one, marine core sea-ice
datapoint (Kageyama et al., 2020). This, alongside the complete loss of summer sea-ice, makes this model of interest for aiding
our understanding of sea-ice loss mechanisms, particularly those related to melt ponds, both for the LIG and for the future
(Guarino et al., 2020b).

Melt ponds are systems of pools that form from meltwater and begin to collect on the Arctic ice surface in spring. They
seem to be critical for understanding HADGEM3-simulated LIG sea-ice loss. Pond-covered ice has a lower albedo, at 0.1–0.5,
than bare ice, at 0.6–0.65, or snow, at 0.84–0.87 (Perovich et al., 2002b; Eicken et al., 2004; Perovich and Tucker, 1997).
Pond-covered ice thus absorbs a higher fraction of incident solar radiation, and transmits a greater fraction of incident radiation
to the ice and ocean below. This difference accelerates the melting of the ice beneath ponds, with melt-rates of pond-covered





ice up to 2-3 times the melt-rate of bare ice (Fetterer and Untersteiner, 1998)). Over the last decades, melt ponds have played
a key role in reducing the surface albedo throughout melt season (Eicken et al., 2004; Maslanik et al., 2007; Perovich et al.,
2007); nearly 60% of the summer sea ice area may be covered by ponds (Fetterer and Untersteiner, 1998; Eicken et al., 2004).

In spite of their importance, melt ponds have only rather recently been explicitly included in CMIP models. In most CMIP6
models, the most common approach is to implicitly parameterise melt ponds by reducing the ice/snow albedo when surface
ice temperatures approach 0 °C (*e.g.* Collins et al., 2006; Curry et al., 2001). This tuning has been relatively successful for
reproducing realistic melt rates for the present day (Collins et al., 2006; Curry et al., 2001). However, pond formation is
affected by sea-ice processes throughout melt season (e.g. evolving topography and snow cover), which this tuning does not
represent (Kwok et al., 2009).

The Taylor and Feltham (2004) melt pond model is one-dimensional, and based on heat and salt balances. The Lüthje et al.
(2006) model simulates the evolution of pond area, and the Skyllingstad et al. (2009) model simulates pond formation on level
ice. The latter two models were combined and improved on by Scott and Feltham (2010), and this new model used to investigate
the impact of snow cover and topography on pond formation. The first physically-based melt pond model for application in a
GCM's sea ice sub-model was developed by Flocco and Feltham (2007) (and improved by Flocco et al. (2012)). Hunke et al.
(2013) implemented an alternative melt pond model into the CICE5 sea-ice model using the fraction of level sea-ice instead of
the ice thickness distribution.

Here, we analyse the first simulation of an ice free Arctic during the LIG using HadGEM3 (Guarino et al., 2020b), with
explicit melt pond dynamics (CICE 5.1), to examine in detail how melt ponds contribute to Arctic sea-ice loss during the
LIG, particularly the enhanced LIG summer sea-ice loss. We investigate possible ice loss drivers, quantifying the impact of
thermodynamic or dynamic processes for the enhanced loss. We then investigate thermodynamic processes in detail, and study
what drives LIG surface albedo changes, and in particular the impact of melt ponds on the LIG-PI albedo difference. Finally, we
study the predictability of summer sea-ice loss from the spring melt pond area, and compare results between the two simulated
periods.

## 2 Methods

### 2.1 The HadGEM3 model

All the simulations analysed in this study use the low resolution version of the latest UK physical climate model, HadGEM3-
GC31-LL, hereafter HadGEM3 (Williams et al., 2018). HadGEM3 is a fully coupled climate model that uses the Unified Model
(UM) (Walters et al., 2017) for the representation of the atmosphere, the Joint UK Land Environment Simulator (JULES) for
the representation of land surface processes (Walters et al., 2017), and the NEMO3.6 (Madec et al., 2015) and the CICE5.1
(Hunke et al., 2015) models for the representation of the ocean and the sea-ice, respectively.

In its low resolution version (N96-ORCA1), HadGEM3 utilizes a horizontal grid-spacing of approximately 135 km on
a regular latitude-longitude grid for the atmosphere. For the ocean, an ortoghonal curvilinear grid with a grid-spacing of
approximately 1° is used. Note that the grid-spacing for the ocean model decreases down to 0.33° between 15° N and 15°



S of the equator, as described by Kuhlbrodt et al. (2018). For the vertical discretization, the UM atmospheric model utilizes 85 pressure levels (terrain-following hybrid height coordinates) while the NEMO ocean model uses 75 depth levels (rescaled-height coordinates).

The modifications and setup of the applied sea-ice model CICE are described in Ridley et al. (2018b). For our study it is important to mention that the albedo calculation is based on the scheme used in the CCSM3 model (Hunke et al., 2015), but includes surface melt ponds by applying the explicit topographic melt pond model of Flocco et al. (2012) and Flocco et al. (2010). Meltwater, formed as a result of snow melt, ice melt and precipitation, runs downhill under the influence of gravity and collects on sea-ice starting at the lowest surface height. The evolution of pond fraction and depth as well as the formation

of ice lids are calculated. In other sea-ice models without an explicit pond scheme, the ice albedo is reduced when the surface temperature approaches freezing temperature (see *e.g.* Hunke et al., 2015) to indirectly account for the impact of melt ponds. This adjustment has been removed here not to double count for the impact of ponds on albedo.

    For full details on model configuration, performance and improved physics compared to older model versions see Williams et al. (2018).

**2.2   Simulations**

The Pre-industrial (PI) simulation used in this study was prepared and run by the UK Met Office as part of the sixth coupled model intercomparison project CMIP6 (Eyring et al., 2016). This simulation uses invariant solar, greenhouse gas (GHGs), ozone, tropospheric aerosol, volcanic and land-use forcing for the year 1850, see Menary et al. (2018) for details. The climate system took about 700 model-years of spin up to attain a steady state. These years are not used in our analysis. Of the subsequent

500 model-years of production run (Menary et al., 2018), the first 200 are used here in our analysis.

    The LIG simulation analysed in this study was first presented by Guarino et al. (2020b); it constitutes the UK's PMIP4 LIG contribution, as part of the wider CMIP6 project. The LIG experiment fully complies with the standard PMIP4 experimental protocol for Last Interglacial climate simulations, as described by Otto-Bliesner et al. (2017). In more detail, this simulation is a time-slice of the Earth's climate at 127,000 years ago (*i.e.* 127k). The Last Interglacial climate was forced using 127 k

constant astronomical parameters based on Berger and Loutre (1991), and constant atmospheric trace GHG concentrations derived from ice core records (see Otto-Bliesner et al. (2017) Table 1 for full details and values used). All other boundary conditions including ice sheets, topography, vegetation, aerosol, volcanic activity, solar constant, *etc.* are identical to the PI simulation.

    The LIG simulation was initialized from the end of the 700 years of PI spin-up. A further 350 model-years of LIG spin-up

were required for the atmosphere and the (upper-) ocean to reach equilibrium. See Williams et al. (2020) for details on how the LIG spin-up was evaluated and what metrics were used to assess the atmospheric and oceanic equilibria. After having attained equilibrium, the simulation was continued for further 200 years of production run. This length of simulation has been shown to be long enough to capture model internal variability (Guarino et al., 2020a).



## 3    Results

### 125    3.1    Enhanced sea-ice loss during the LIG

Here, we examine the HADGEM3-simulated LIG summer sea-ice loss, in order to identify contributing factors. Figure 1 compares the annual cycle of the Arctic sea-ice area between the PI and LIG period from the HadGEM3 simulations.

In the HadGEM3 simulations the LIG winter ice area is slightly lower than the PI, with the smallest difference in area at $0.61\pm0.56$ million $km^2$ in early April (Fig. 1). Compared to the PI, an enhanced rate of LIG sea-ice melt from early May until
late June leads to a consistently ice-free LIG Arctic from early August until early October (Figs 1 and 2). The minimum LIG ice area in September is $0.09 \pm 0.11$ million $km^2$, while the PI minimum ice area is $5.54 \pm 0.98$ million $km^2$. Long-term mean ice thickness during the LIG is also thinner than during the PI in all months (not shown).

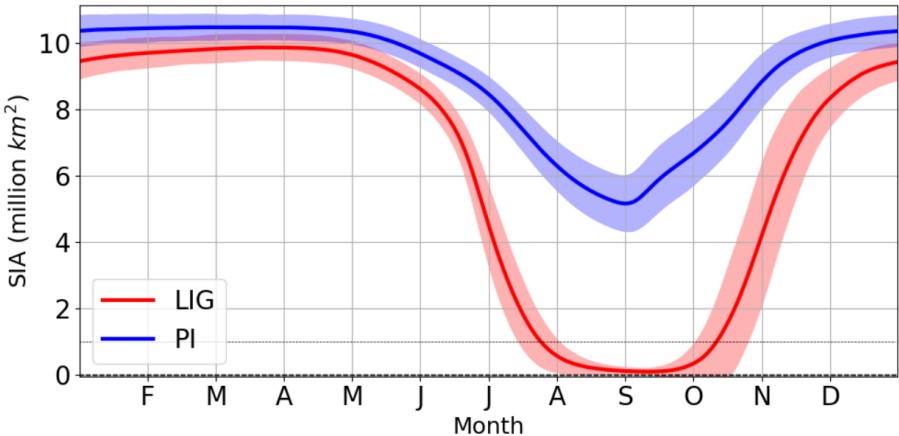

**Figure 1. Annual cycle of Arctic sea-ice area (SIA).** The 200-year mean calculated from daily model output from 70°N - 90°N for the LIG (red) and PI (blue) simulations is shown. Shaded area is $\pm$ twice the standard deviation (inter-annual variability.

The earlier sea-ice retreat results in a warmer and less salty ocean surface during July, August and September during LIG (Figs. 3 and 4). The differences reach more than 5 °C (LIG-PI SST anomaly) and around 1-2ppt. The interaction of ocean
surface conditions and sea-ice extent in the model simulations potentially allow SST observations from proxy records to be used as a signature of ice conditions during LIG. In particular it may be informative for the marine core community to search for large summer SST differences from the PI, of $> 4$ °C, at latitudes near the expected LIG ice edge (e.g. 70 °N; east of Greenland) and smaller summer SST differences from the PI, of $>2$°C, in the central Arctic/Beaufort sea. This could help identify an sea-ice free summer LIG Arctic in marine observations. However, we note that a partially ice-covered summer Arctic, with thin
LIG sea-ice, might have a similar signature to this but with lower magnitude temperature anomalies everywhere, as very thin summer ice cover has a small insulating effect. Thus it is additionally useful to consider SSS changes (Fig. 4). These have a clearer signature than SST. The complete loss of summer LIG Arctic ice causes the region with LIG winter ice cover to become significantly fresher, than the PI, in the summer (around 1-2ppt in June and July), additionally a difference of 0.5-



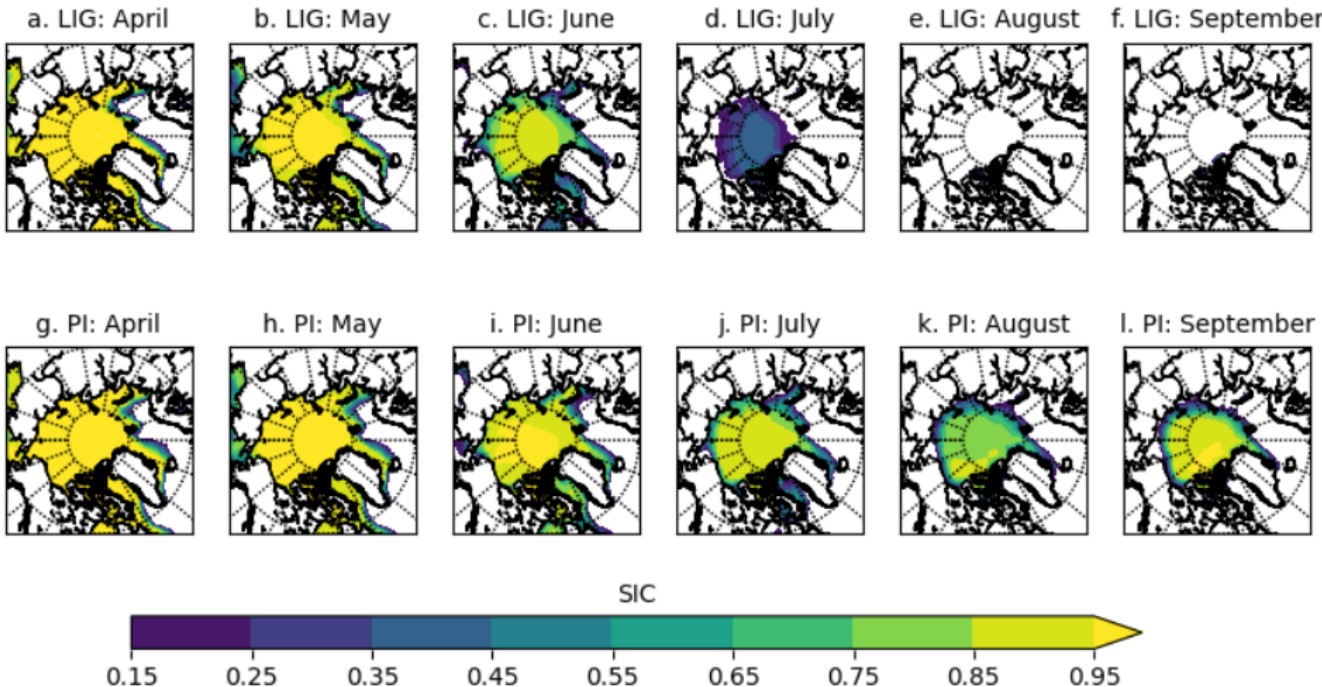

**Figure 2. Maps of mean simulated sea-ice concentration for LIG and PI.** The 200-year mean over the Northern Hemisphere for each month from April-September shown is computed as the time-average from monthly model output. Only cells with long-term mean SIC>0.15 are shown.

1.5ppt is retained at least from March-September. Beyond helping identify the observational signature of a sea-ice free Arctic,
we of course wish to know: what causes the large differences in summer sea-ice in this model? Particularly, why does the spring melt increase so significantly in spite of the lower atmospheric $CO_2$ concentration in the LIG?

Processes with a significant impact on summer sea-ice include: thermodynamic processes such as ice-albedo feedbacks, ocean heat transport, and cloud cover feedbacks. The preconditioning of winter sea-ice may also lead to a reduced sea-ice extent the following summer (Parkinson and Comiso, 2013; Williams et al., 2016). In addition, summer sea-ice extent may
be affected by changes to sea-ice distribution (caused by changes to wind patterns and ocean circulation (Kageyama et al., 2020; Deser and Teng, 2008; Wang et al., 2009); further discussed in Section 3.1.1). But which of these processes is is most significant for the enhanced LIG sea-ice loss compared to the PI?

We first look at the spin-up simulation for the LIG, with a particular focus on the first year of the spin-up (Figs. 5 and 6). From Fig. 5: the winter Arctic sea-ice retains a similar area to the PI control period over the first 15 years of the spin-up
period, beyond which it decreases only slightly (<1 million km $^2$) from the PI control. However, August sea-ice area halves during the first year of the spin-up run from nearly 6 million to around 3 million km $^2$ and an ice-free summer state is reached after only 4 years. This halving of August sea-ice area during the first year of spin-up shows that preconditioning does not



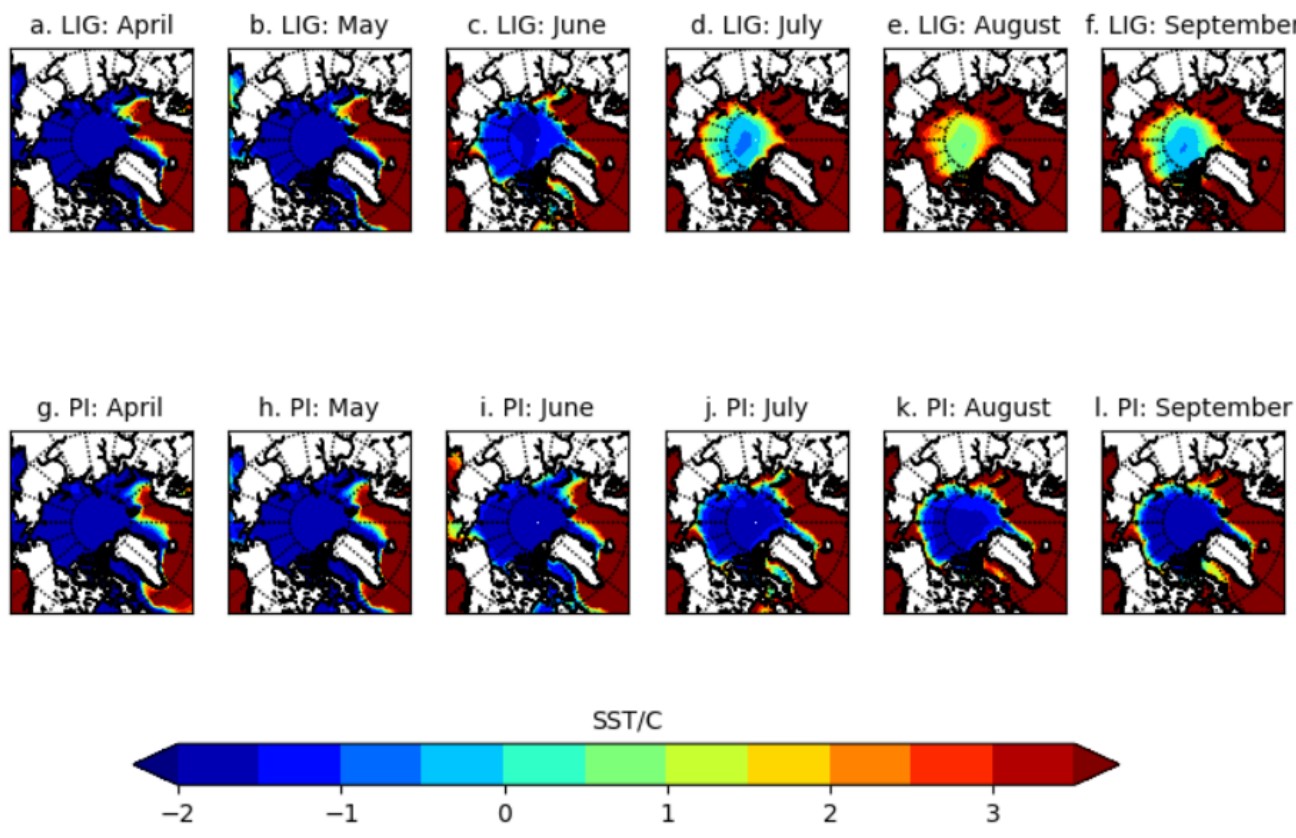

**Figure 3. Mean sea surface temperature °C, for the LIG and PI.** The 200-year mean over the Northern Hemisphere for each month from April-September is shown, computed as the time-average from monthly model output.

play the dominant role in the enhanced melting of sea-ice during the LIG spin-up. Similarly, upper ocean heat transport, often linked to reduced sea-ice (Holland et al., 2006; Steele et al., 2010), takes decades to equilibrate in numerical models after a

significant perturbation, and deeper ocean heat transport, centuries to millenia (Kantha and Clayson, 2000), suggesting ocean heat transport is not the first-order driver of the rapid summer ice loss observed in the spin-up. Therefore, other factors that are present by the summer of LIG spin-up year 1 must be key to the enhanced melt throughout the simulated LIG control period. In order to deduce the importance of two of the remaining processes, ice-albedo and cloud cover feedbacks, we consider the surface energy budget (Fig. 6). This is because the Arctic Ocean's surface heat balance is closely linked to its ice mass balance

(Uttal et al., 2002; Untersteiner, 1961): at the surface, shortwave flux in particular is known to play a dominant role in summer sea-ice melt, and is linked to ice-albedo feedbacks (Maykut and Perovich, 1987); longwave flux is related to the longwave cloud radiative forcing, and cloud-cover feedbacks (Gupta et al., 1992; Niemelä et al., 2001). Summer positive TOA radiation leads to a positive net shortwave (SW) flux anomaly, of up to 75 Wm$^{-2}$ in June, that contributes nearly all of the net downwards surface heat flux anomaly (Fig. 6(a)). Unlike changes related to preconditioning and ocean heat transport, this net SW flux anomaly is







**Figure 4. Mean sea surface salinity (in parts per thousand, ppt), for the LIG and PI.** The 200-year mean over the Northern Hemisphere for each month from April-September is shown, computed as the time-average from monthly model output.

already present in the first year of LIG spin-up run, reaching 55 $\mathrm{Wm}^{-2}$ in June (Fig. 6(b)). This immediate response, coupled with the immediate halving of August sea-ice area, suggests the absorbed shortwave radiation plays a key role in the observed summer ice loss , and thus that ice-albedo feedbacks may also be important. We note also that the longwave anomaly accounts for $< 5$ $\mathrm{Wm}^{-2}$ of the total surface heat flux anomaly in Fig. 6(b), so longwave forcings and feedbacks related to cloud cover are not dominant contributors to the enhanced LIG sea-ice loss. Other differences between the LIG and PI surface heat budget contribute $< 20$ $\mathrm{Wm}^{-2}$ monthly to the surface energy budget (Fig. 6). This shows that thermodynamic processes that led to the enhanced LIG summer sea-ice melt must predominantly result from this surface SW anomaly.


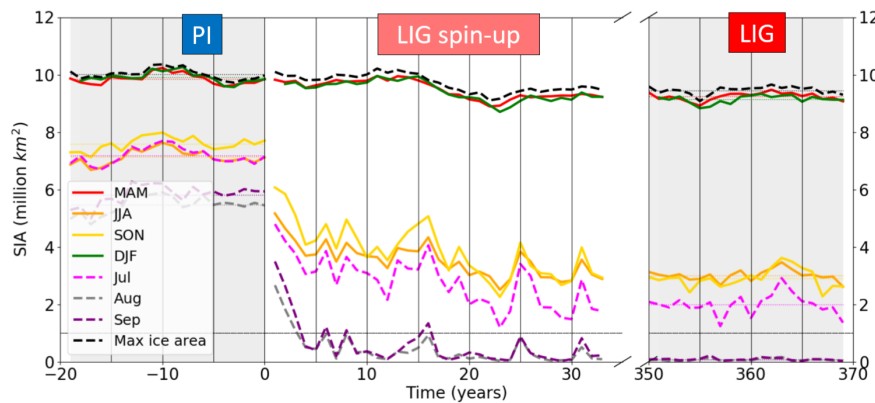

**Figure 5. The loss of Arctic sea-ice during the LIG spin-up.** Comparison of seasonal Arctic sea-ice area (SIA) between simulated LIG and PI and first 35 years of LIG spin-up period, over the region 70°N - 90°N, computed from monthly model output using ice-covered grid-cells (SIC>0.15) only. The black dashed line corresponds to maximum SIA reached any month during the year. A new ice-free state in August and September is reached within the first 5 years of the LIG spin-up.

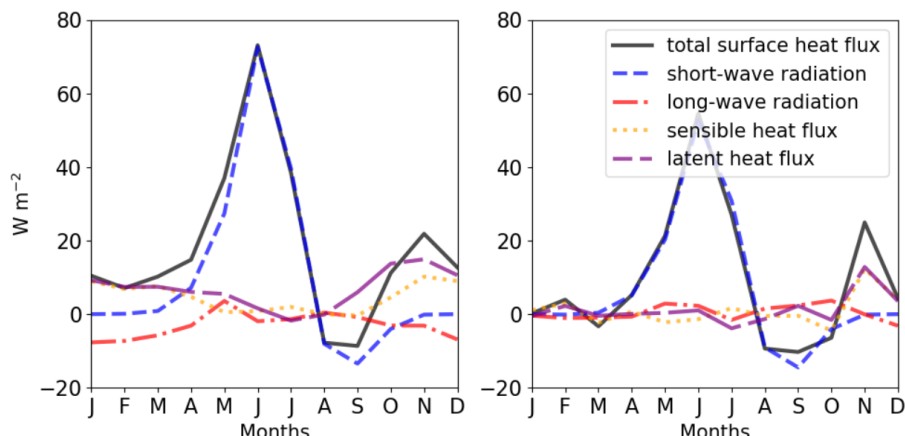

**Figure 6. LIG-PI anomaly of the simulated Arctic surface energy budget from the PI simulation, for (a) the LIG simulation (adapted from Guarino et al., 2020b) and (b) the first year of the LIG spin-up period.** For the LIG, PI, and the first year of the LIG spin-up period, the spatial average was computed from monthly data over the region from 70°N - 90°N for the simulated short-wave radiation, long-wave radiation, sensible heat flux, and latent heat flux. Anomalies are found from the long-term means. The total surface heat flux anomaly (black) is the sum of these four heat budget anomalies.

Therefore, by elimination, the dominant contributors to the enhanced LIG summer sea-ice loss are (i) thermodynamic processes, driven by this surface SW anomaly and likely related to ice-albedo feedbacks, and/or (ii) changes to the summer ice distribution. In the next section, we investigate the relative importance of these processes.



### 3.1.1 Thermodynamic versus dynamic processes

Sea-ice increase and loss are driven by a combination of 'thermodynamic processes', which involve ice-atmosphere and ice-ocean heat fluxes, and 'dynamic processes', which involve changes to the local ice volume due to convergent or divergent ice motion (Li et al., 2014) caused by wind or ocean stress (Goosse and Fichefet, 1999). The key driver of ice motion is the wind stress forcing (Köberle and Gerdes, 2003): present-day interannual variability in Arctic summer sea-ice extent is linked to changes to sea-ice distribution caused by wind pattern variability due, for example, to large-scale atmospheric variability (Deser and Teng, 2008; Wang et al., 2009). In order to determine to what extent the enhanced LIG sea-ice loss is caused by these dynamic processes, or by thermodynamic processes related to the SW anomaly shown in Fig. 6, we examine the ice volume tendencies due to dynamics and due to thermodynamics (shown respectively in Fig. 8 and Fig. 7). The largest

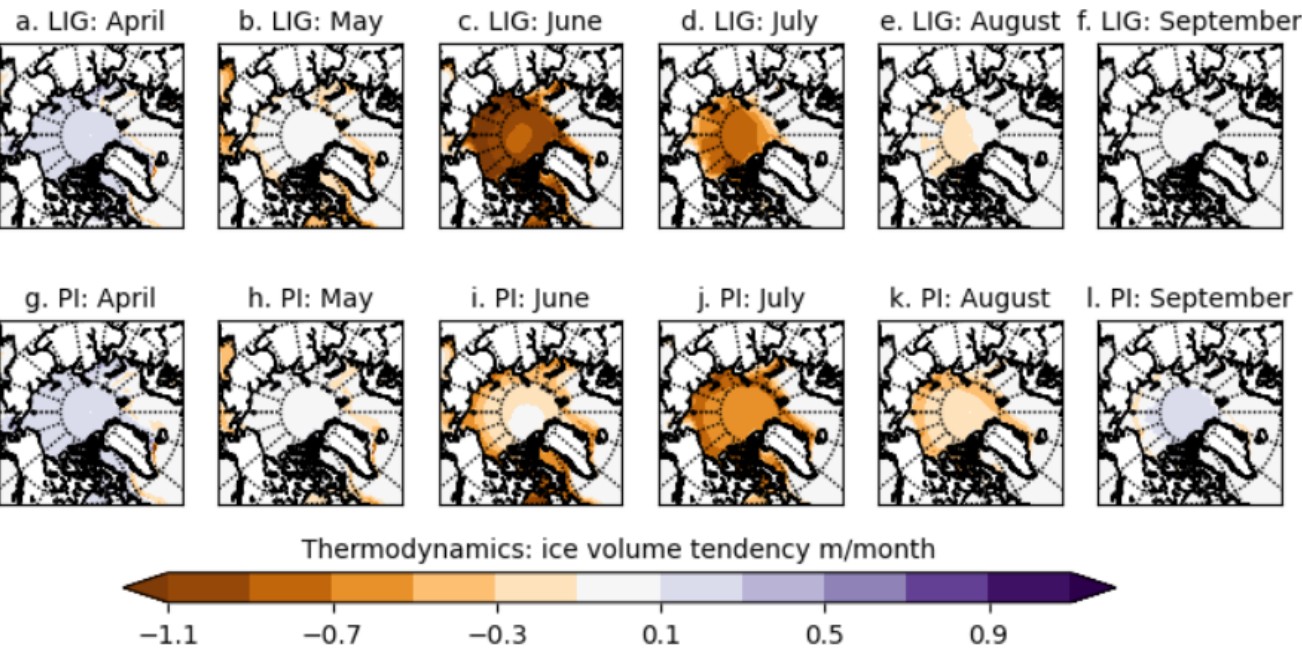

**Figure 7. Maps of mean simulated ice volume tendency [m/month] due to thermodynamics, for the LIG and PI.** The 200-year mean over the Northern Hemisphere for each month from April-September is shown, computed as the time-average from monthly model output.

difference occurs in June with 70 to 110 cm/month of ice melt during the LIG, in contrast to the 10 to 30 cm/month ice melt in the PI simulation (Fig. 7). The thickness changes caused by dynamic processes (Fig. 8) are smaller: note that different scales are used in the maps for figure clarity. While a divergent ice drift reduces ice thickness during most months in the PI simulation, this is not the case during the LIG period. However, the differences in magnitude between the two periods are generally less than 10 cm/month. This demonstrates that, during spring and summer, changes in thermodynamic rather than changes in dynamic processes are the first-order driver causing increased ice melt for the LIG simulation. Thus, the surface





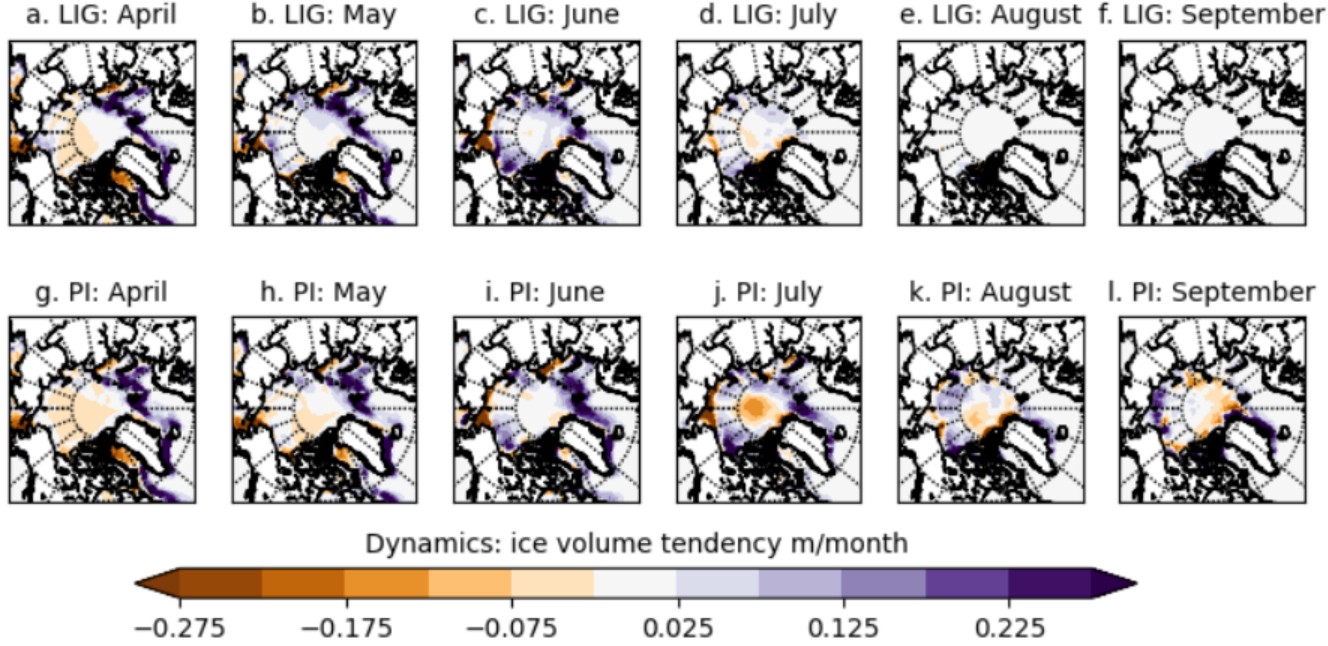

**Figure 8. Maps of mean simulated ice volume tendency [m/month] due to dynamics, for the LIG and PI.** The 200-year mean over the Northern Hemisphere for each month from April-September is shown, computed as the time-average from monthly model output. Note: as the magnitude of the change in ice volume due to dynamics during the summer months is significantly smaller than the change due to thermodynamics, the colorbar scale is 1/4 of the colorbar scale shown in Fig. 7.

SW anomaly shown in Fig. 6 is key to the enhanced LIG ice loss. Therefore, we investigate in the next section the source of this surface net SW anomaly.

### 3.1.2 Melt ponds and albedo feedback

At the ice surface, 20-30% (Perovich et al., 2002a) of incoming SW radiation is directly absorbed. Absorbed radiation may be transmitted through the ice to the ocean, which warms, leading to processes including further sea-ice melt (Perovich et al., 2002a). The ratio between incoming and absorbed SW radiation is determined by the surface albedo. Sea-ice has a high albedo and thus tends to predominantly insulate the ocean below it by reflecting most of the incoming SW radiation. This can be seen from the similar spring SST under the PI and LIG Arctic sea-ice (Fig. 3). Therefore, the surface net SW anomaly results from the anomaly of the surface incoming SW flux (caused by the TOA anomaly), and may be amplified (or reduced) by ice-albedo feedbacks that differ between the PI and LIG (Fig. 6). In this section, we investigate incoming surface SW radiation, and its amplification by ice-albedo feedbacks.

We first consider the LIG-PI anomaly of the surface albedo in the Arctic Ocean. This is small in April, but grows throughout summer (Fig. 9): the LIG-PI surface albedo anomaly of <-5% in April grows to -35% by July. Formation of melt ponds (which





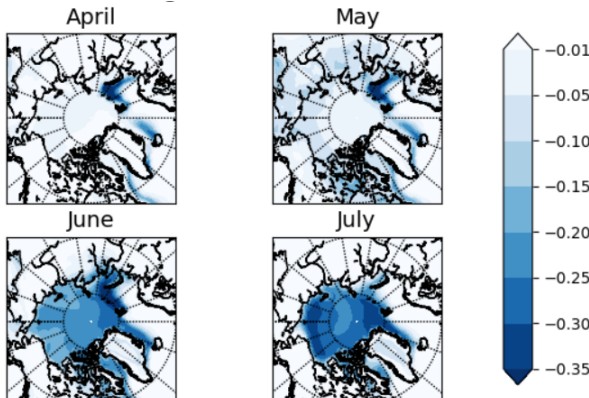

**Figure 9. The LIG-PI albedo difference.** For both LIG and PI, for each grid-cell, from monthly model output, for each month the albedo was computed as *1-(absorbed surface short-wave flux/downwelling surface short-wave flux)*, where absorbed and downwelling short-wave flux are model output variables. The LIG-PI anomaly is computed from the 200-year time averages for the PI and LIG.

have lower albedos that sea-ice) contributes to the albedo feedback effect (Perovich et al., 2002a; Hunke et al., 2015), so we evaluate the magnitude of the changes of pond evolution. We look at the mean first day each grid-cell has pond fraction greater
than 1% as a proxy for melt onset (which was not available in the model output). The geographical pattern of first day of melt pond formation is similar for both periods, with ponds forming first at low latitudes and gradually spreading to higher latitudes (Fig. 10). Pond formation begins on average 8.4 days earlier during the LIG compared to the PI. The difference between the two periods increases as the latitude increases. Ponds south of 70°N (and distant from the sea-ice edge) begin to form only 0-5 days earlier during the LIG than the PI. Ponds begin to form at higher latitudes 15-20 days earlier during the LIG (Fig. 10(c)).
As well as forming earlier in the LIG compared to the PI simulation, ponds also cover a greater area fraction of the ice-covered grid-cells throughout the spring. For both periods, the long-term mean of the pond fraction of the grid-cell for each month is greater for the LIG than the PI from May-June, but smaller from July (Fig. 11). This is simply because melt ponds cover a larger area of the ice for the LIG than the PI in spring to early summer, but very little ice remains for the LIG simulation from July onwards, so there is less sea-ice available for ponds to cover.  At the end of April, pond formation becomes stronger during
the LIG than the PI. From early May, the total Arctic pond area increases exponentially with rate of $\sim 0.14$ day$^{-1}$, until late May when the rate of pond formation begins to decelerate; maximum pond area at $2.67\pm0.37$ million km$^2$ is reached mid-June (Fig. 12). For the PI, from early May, pond area increases exponentially with rate constant $\sim 0.09$ day$^{-1}$, until early June when the rate of pond formation decelerates until the maximum pond area is reached at $2.54\pm0.45$ million km$^2$ in mid-July. A consistently higher fraction of the LIG sea-ice is pond-covered throughout the spring, with a peak value of $45.3\pm3.7$ %
(in early July) compared to peak value of $34.4\pm5.1$ % for the PI (in late July - not shown). Note, only ice-covered grid-cells (defined as cells with SIC > 0.15) above 70 °N were used to calculate the 200-year mean and standard deviation for each day of the year for Fig. 12 (and subsequent figures where indicated in the caption).




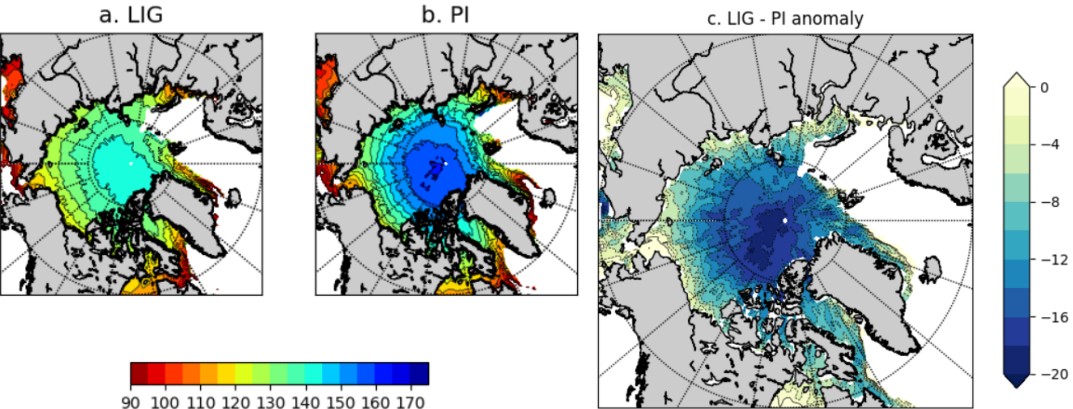

**Figure 10. First day of melt pond formation as proxy for melt onset.** The mean day of the year the grid-cell melt pond fraction grew above 0.01 for (a) LIG, (b) PI, and (c) the LIG - PI anomaly. All figures show the first 50-year mean, from daily model output. Only regions still ice-covered (SIC>0.15) when melt ponds began to form are shown.

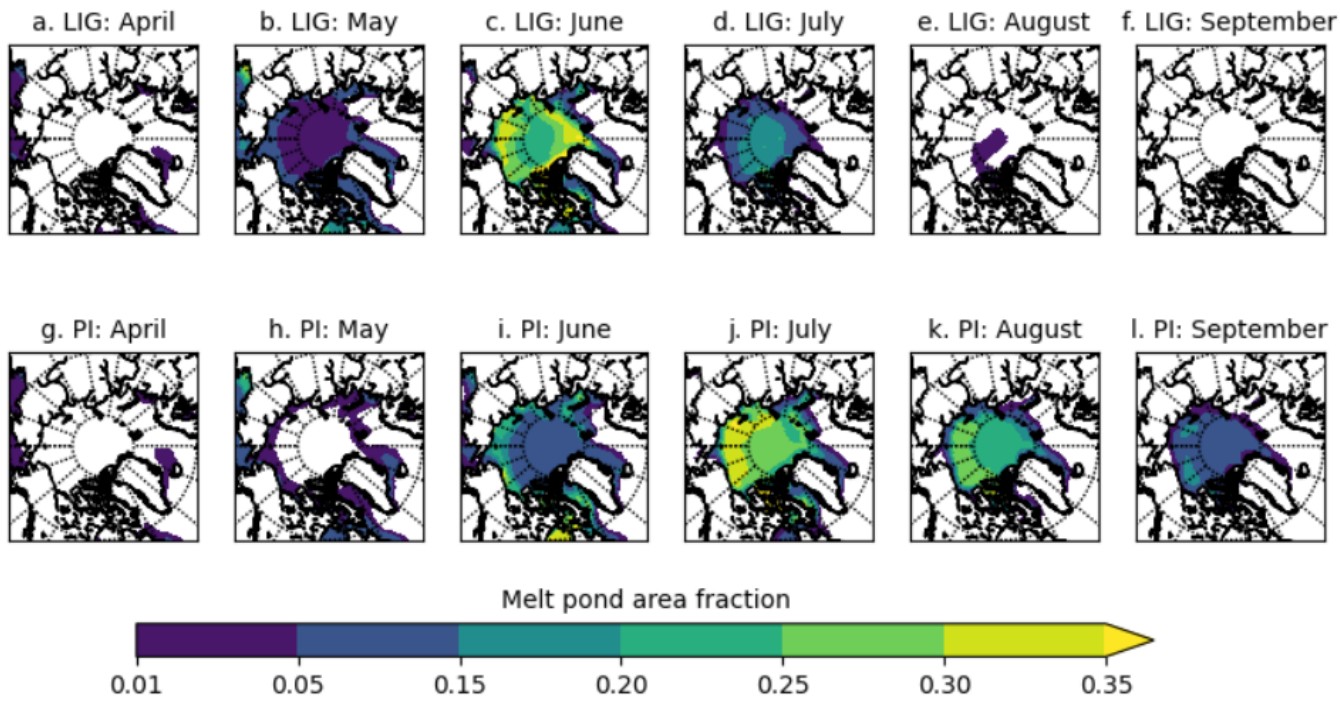

**Figure 11. Melt pond fraction.** Monthly simulated melt pond fraction of the grid-cell for LIG and PI. The 200-year mean over the Northern Hemisphere for each month from April-September shown is the time-average from monthly model output.





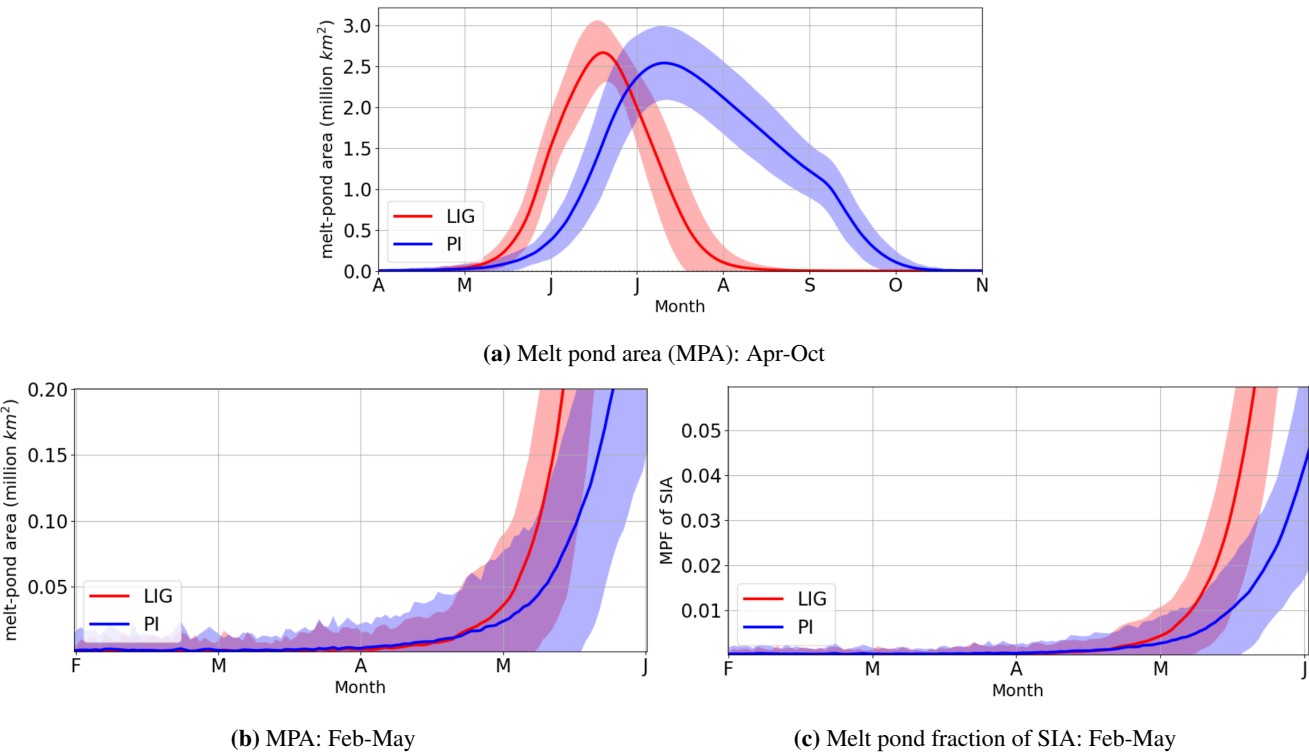

**(a)** Melt pond area (MPA): Apr-Oct

**(b)** MPA: Feb-May

**(c)** Melt pond fraction of SIA: Feb-May

**Figure 12. Annual cycle of Arctic melt pond area.** (a) and (b) show total melt pond area, (c) shows melt pond area of sea-ice. For each figure, for the LIG (red) and PI (blue) simulations, the 200-year mean was computed from daily model output variables grid-cell sea-ice area and melt pond area, over the region from 70°N - 90°N. Shaded area is $\pm$ twice the standard deviation. Only ice-covered grid-cells (SIC > 0.15) were used.

In order to (i) compare incoming SW radiation between the two time periods (ii) quantify the importance of the albedo difference (Fig. 9) and (iii) characterise to what extent melt pond formation (Fig. 11) causes this albedo difference, we compare the

total incoming (or 'downwelling') and net (or 'total absorbed') SW radiation over ice-covered grid-cells. To directly compare the same ice-covered geographical region between the PI and LIG simulations, for any given day of the year, all quantities are computed using only cells that had, on this day of the year in both simulations, long-term mean SIC>0.15.The net absorbed SW radiation was further broken down into the fraction absorbed by open water, by melt ponds, and by bare ice and snow, using their respective area fractions of the grid-cell and estimating their respective albedos: for every grid-cell for every day

of model data, the cell albedo (reflected SW flux/downwelling SW flux) was calculated. An open-water and pond albedo of 0.07 and 0.27, respectively, alongside the area fraction of these two components for each cell, allows the proportion of SW flux absorbed by each of these two components to be computed. The remainder of SW flux absorbed is attributed to exposed ice and snow. Albedo changes can be compared between the LIG and PI. For both time periods, North of the Equator, from January to June, the TOA downwelling SW flux increases. This increases the surface downwelling SW flux, which leads to an

increase of the absorbed SW flux. Fig. 13(a) shows these changes in surface SW flux for the LIG: over ice-covered cells, an





**(a)** LIG: estimated flux absorbed

**(b)** LIG - PI anomaly: estimated flux absorbed

**(c)** LIG - PI anomaly: energy absorbed from 1st May up to this day

**(d)** Ice-covered region

**Figure 13. Incident surface short-wave (SW) flux absorbed by each surface type.** Calculated on ice-covered grid-cells. (a) the LIG simulation (b) anomaly of the LIG from the PI simulation, (c) anomaly of the LIG from the PI simulation of cumulative SW energy/unit area absorbed on ice-covered cells by each each surface type (d) ice-covered grid-cells for four example days of the year in blue. Figures show quantities calculated from the mean daily downwelling (yellow) and total absorbed (gold) SW flux at the surface, and approximate breakdown of SW flux absorbed by open water (blue), exposed ice and snow (green), and melt ponds (black), all computed from daily model output from the first 50 years from each of the PI and LIG simulations. Further details in Section 3.1.2





increase of downwelling flux from $99 \pm 6$ Wm$^{-2}$ on April 1st, to $313 \pm 25$ Wm$^{-2}$ on June 15th, leads to the absorbed flux at the surface increasing from $26 \pm 2$ Wm$^{-2}$ to $201 \pm 19$ Wm$^{-2}$. Thus, from April 1st to June 15th, the LIG surface albedo decreases by 38% (from 74% to 36%). Similar computations for the PI yield an albedo decrease of only 17%. Thus, albedo changes were more significant for the LIG. In May, the increasing gradient of the surface net SW flux, despite the decreasing

gradient of the surface downwelling SW flux, is caused by ice-albedo feedbacks.

Using Fig 13(b), we compare incoming SW radiation between LIG and PI, and use this to quantify to what extent ice-albedo feedbacks modify the surface net SW anomaly shown in Fig. 6. Due to the TOA LIG-PI SW anomaly outlined in 1, the anomaly of the surface downwelling SW flux increases from 4 Wm$^{-2}$ on April 1st to 36 Wm$^{-2}$ on June 15th. This increases the anomaly of the absorbed SW radiation at the surface from 2 to 85 Wm$^{-2}$, as seen from Fig. 13(b) In Fig 13(b), it is remarkable that

the surface absorbed anomaly is 2.0 times the downwelling anomaly by May 24th, and 3.3 times the downwelling anomaly by June 7th: the difference in surface albedo between the LIG and PI causes the difference in downwelling radiation to be amplified threefold. We have demonstrated that the surface net SW anomaly in Fig. 6(a) was caused by the TOA SW anomaly, significantly amplified by stronger LIG than PI ice-albedo feedbacks.

Thus, for both time periods, from spring to summer: the radiative forcing triggers the albedo feedback, and the increase in the

radiative forcing continues to strengthen this feedback into the summer. However, for the LIG, the stronger radiative forcing amplifies the albedo feedback more significantly, so that a much greater fraction of the incoming SW radiation is absorbed in the Arctic. This significantly changes the Arctic heat budget and ultimately results in a complete loss of Arctic sea-ice by August.

Ice-albedo feedbacks result from changes in ice cover and pond formation. To determine to what extent melt pond formation

(Fig. 11) impacts the albedo anomaly shown in Fig. 9, and causes the surface net SW anomaly, we consider the LIG-PI anomaly of the SW radiation absorbed by each surface type. From Fig. 13(b): the melt pond anomaly is comparable to the open water anomaly from May to June, and much greater than the ice and snow anomaly. In particular, from May 15th up until June 20th, just as the LIG-PI anomaly of the downwelling surface SW flux is reaching its peak, the magnitude of the melt pond anomaly is at least 0.5 times the open water anomaly (and at least 1.3 times the ice anomaly). The melt pond anomaly from May 23rd

(at 15 Wm$^{-2}$) to June 4th (at 74 Wm$^{-2}$) is greater than both the open water, and the ice and snow, anomalies. Therefore, the role of melt ponds in decreasing the LIG albedo, thus amplifying the surface net SW flux anomaly, is particularly significant as the surface downwelling SW flux anomaly grows through May and peaks in June.

The cumulative SW energy absorbed over melt season by ponds, and by open water, may be computed from Fig. 13. We consider the LIG-PI anomaly of the energy absorbed from May 1st onwards (approximately the beginning of melt season),

calculated from Fig. 13(b) and shown in Fig. 13(c) Over ice-covered cells, by May 25th, the anomaly of the energy absorbed by ponds (12 MJ m$^{-2}$) is half that of open water (24 MJ m$^{-2}$). By June 11th, the anomaly of the energy absorbed by ponds and by open water is equal, at 55 MJ m$^{-2}$.

This demonstrates the significant impact of melt ponds on the surface energy balance, and by extension their key role in enhancing LIG summer sea-ice loss.





## 3.2 Melt ponds and sea-ice predictability

Today's diminishing Arctic sea-ice has led to a new focus on seasonal forecasting of Arctic sea-ice conditions, and especially predicting the minimum sea-ice area each year (Williams et al., 2016). Spring melt ponds are a good predictor for summer sea-ice conditions (Schröder et al., 2014). It is of interest to see how the predictability of spring melt pond area and August-October sea-ice area varies between the PI and LIG, since this may yield insight into how predictability may change in future under conditions of reduced Arctic sea-ice. Predictability is investigated here by considering interannual variability within each of the LIG and PI periods. The interannual variability in the radiative forcing at the surface is much smaller than the difference in the radiative forcing between the PI and LIG. Thus, investigating the relation between spring melt pond area and August-October sea-ice area within each of these two periods gives insight into the impact of melt ponds on the summer sea-ice area for similar TOA radiative forcing and winter ice conditions each year.

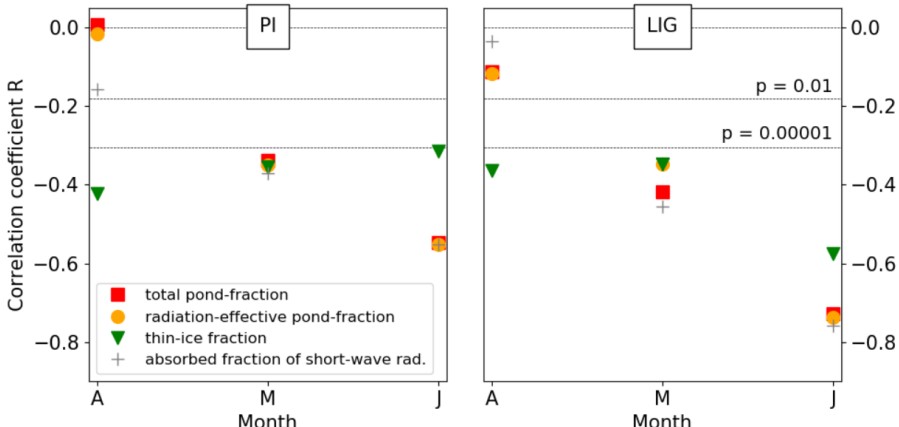

**Figure 14. Correlation between spring melt pond fraction (MPF) of sea-ice area (SIA) and autumn SIA for the (a) PI, and (b) LIG.** For both (a) and (b), 200 years of monthly model data were used over the region 70°N - 90°N. For mean MPF of SIA in April, May and June, Pearson's correlation coefficient R was calculated between melt pond area and mean August-October SIA the same year (shown in red). For comparison, R was calculated also between mean ASO SIA and each of: mean radiation-effective MPF of SIA (gold), mean fraction of incident short-wave radiation absorbed (grey), and mean thin-ice fraction of ice area (green; defined as fraction of ice-covered grid-cells with ice thickness≤1.4m)

For the 200 years' of simulation output for both LIG and PI, Pearson's correlation coefficient between the mean August to October sea-ice area, and the mean monthly melt pond fraction of sea-ice in April, May and June, was calculated (Fig. 14). For comparison, the same computation of the correlation was carried out for August-October sea-ice area and April, May and June thin-ice fraction of sea-ice, as the ice state in spring is known to affect the summer sea-ice area (Schröder et al., 2014). Also for comparison, the correlation between sea-ice area and fraction of incoming short-wave radiation absorbed was found, as this fraction over ice-covered cells accounts for albedo changes from open water as well as ponds in these cells. The same





computation was also carried out for the 'radiation-effective pond fraction', denoting the fraction of grid cell area covered by ponds that are not covered by an ice lid and thus are expected to affect the surface albedo (DuVivier, 2018). By May, for both simulated periods, there is a significant ($p < 10^{-5}$) negative correlation between melt pond formation and summer sea-ice area (MPF correlation), of -0.34 for the PI and -0.42 for the LIG. This is only a slightly weaker correlation than that between

absorbed short-wave fraction and summer sea-ice area (SWF correlation), of -0.37 for the PI and -0.46 for the LIG (Fig. 14). By June both of these correlations were much stronger than the correlation between thin-ice fraction and summer sea-ice area: the MPF correlation is -0.55 for the PI and -0.73 for the LIG, and the SWF correlation is -0.55 for the PI and -0.76 for the LIG. Compared to the PI, the LIG April sea-ice area above 70 °N is on the order of a million km$^2$ lower, and the mean sea-ice thickness is about 1m less, so we demonstrate that the seasonal predictions for summer sea-ice improves for thinner sea-ice as

it is more sensitive to pond formation and albedo changes.

### 3.3 How does HadGEM3 compare with other models?

Despite their significant impact on sea-ice melt, melt ponds have only recently been explicitly included in global climate models (*e.g.* Collins et al., 2006; Curry et al., 2001). Prior to this, models without an explicit pond scheme had the ice albedo reduced when conditions approached near-melting (Hunke et al., 2015). Until now, it has not been clear whether this melt pond

parameterisation approach adequately accounts for the impact of ponds during warmer-than-present day conditions (Kwok et al., 2009).

Of the CMIP6 models that have simulated the LIG according to CMIP6-PMIP4 experimental protocol, the two to include an explicit melt pond scheme are the two with the lowest mean September sea-ice extent: HADGEM3, 0.07 million km$^2$, and CESM2, 2.55 million km$^2$. The only other model that simulates a September LIG sea-ice area less than 4.0 million km$^2$ is

NESM3, at 2.96 million km$^2$, but this model significantly overestimates the amplitude of the seasonal cycle: it overestimates LIG winter SIE by two standard deviations from the multi-model mean, by far the most significant outlier among the models. This suggests NESM3's representation of sea-ice is unusual; as such, we regard it as unlikely that the low LIG sea-ice in this model is related to its representation of melt ponds. The multi-model mean of September SIE of the remaining models, which all use parameterised rather than explicitly modelled ponds, is $5.25 \pm 2.20$ million km$^2$ (Kageyama et al., 2020). Since only 12

models have run this simulation, it is not possible to say that implementing an explicit melt pond scheme will tend to result in lower simulated LIG summer ice area; other aspects of the model simulation set-up may also be significant in the simulation of Arctic sea-ice. For example, the IPSL-CM6L model simulates some compensating ocean circulation and cloud cover changes that contribute to preserving LIG summer sea-ice (Kageyama et al., 2020).

However, despite these caveats, it is striking that the only two models to include explicitly modelled melt ponds also simulate

minimum LIG SIE at least 2 million km$^2$ less than the multi-model mean of the remaining models, whilst simulating maximum LIG SIE (and the PI SIE annual cycle) similar to the multi-model mean.





## 4 Conclusions

In terms of understanding mechanisms driving possible Arctic sea-ice change during the LIG, Guarino et al. (2020b) indicated that albedo changes are highly influential. Here we provide a more in-depth follow-on study which analyses the potential thermodynamic and dynamic contributors, and provides multi-model context that was not available to Guarino et al. (2020b). This addresses the key gaps in our understanding of why melt pond schemes matter during warm climates, and specifically answers the question of how HadGEM3's melt pond scheme contributes to the simulated ice loss in the Arctic during the LIG.

We identify the key thermodynamic processes that lead to the ice-free summer LIG Arctic as albedo feedbacks triggered by the radiative forcing. The TOA SW positive downwelling LIG-PI radiation anomaly in the spring and summer is halved by the time it reaches the surface (Guarino et al., 2020a; Kageyama et al., 2020). However, we show that the greatly strengthened albedo feedback in the LIG compared to the PI (with LIG-PI surface albedo anomaly of <5% in April, growing to 30-35% by July) re-amplifies this small surface anomaly by the summer by up to a factor of three. This means up to 80Wm$^{-2}$ more short-wave radiation is absorbed at the surface, leading to significant differences between the LIG and PI surface heat budgets, and explaining the greatly enhanced LIG sea-ice melt compared to the PI. We further demonstrate that ponds play a key role in reducing the albedo and strengthening the feedback process: through May and June, the downwelling surface SW flux anomaly peaks, and over ice-covered cells, melt ponds and open water account for a similar proportion of the surface absorbed anomaly. Therefore explicitly modelled melt ponds contribute significantly to the simulated loss of summer sea-ice.

Our analysis of ice volume tendencies demonstrates the difference in HadGEM3-simulated LIG and PI summer ice melt rates is predominantly driven by thermodynamic processes. This is interesting since atmospheric and ocean circulation changes often contribute to ice loss (Li et al., 2014; Köberle and Gerdes, 2003; Kageyama et al., 2020; Goosse and Fichefet, 1999); however this is not the case here. We find that for both the PI and LIG dynamic processes, driven by wind and ocean stress, lead to less than 10cm/month of Arctic ice volume change through the spring and summer months. By contrast, thermodynamic processes result in most ice volume change for both simulations, and account for the enhanced LIG ice loss. Three to five times more LIG sea-ice is lost than during the PI by June, in most Arctic regions: 70-110 cm/month of LIG sea-ice is lost due to thermodynamic processes, compared to 10-30 cm/month during the PI.

Given today's new focus on seasonal forecasting of Arctic sea-ice conditions, and especially predicting the minimum sea-ice area each year (Williams et al., 2016); we also investigated whether melt ponds are a good predictor for summer sea-ice conditions (Schröder et al., 2014). Strong correlations between April-June melt pond area and August-October sea-ice fraction show that explicitly modelling melt ponds significantly impacts summer sea-ice area. Much stronger correlations were found for the LIG, which has ice that is thinner and thus more sensitive to melt pond formation than the PI. This is of concern for future seasonal sea-ice predictions: as Arctic ice continues to thin, the spring melt pond area each year may be an increasingly important and reliable indicator of the September sea-ice area.

In conclusion, whilst both models with both parameterised and explicitly modelled melt ponds are relatively successful in representing present-day sea-ice behaviour (Collins et al., 2006; Curry et al., 2001; Flocco et al., 2012), we find that they likely simulate significantly different sea-ice behaviour under forcings other than the present-day. Multi-model context, alongside

our new analysis above, suggests that a better representation of the contribution of melt ponds to enhanced Arctic sea-ice melt during the Last Interglacial is important. The relatively close match of HadGEM3 surface air temperatures to those derived from proxy records (Guarino et al., 2020b), and expected match of CESM2 (see figures in Otto-Bliesner et al. (2020)), suggest that explicitly modelled pond formation for the LIG period does appear to be crucial to simulate realistic areas of summer

sea-ice and Arctic temperature changes in current CMIP models. This is highly relevant to future projections of sea-ice loss, particularly when predicting the Arctic amplification of anthropogenic forcing; this requires accurate representation of albedo feedback mechanisms (Smith et al., 2019; Stuecker et al., 2018). Thus, our study of HadGEM3 supports the idea that an explicit, realistic melt pond scheme is required for both past and future sea-ice and climate projections.

*Data availability.* The HadGEM3 model outputs prepared for CMIP6, including the simulated PI, are in the ESGF archive: https://doi.

org/10.22033/ESGF/CMIP6.419 (Ridley et al., 2018a). Processed and additional HadGEM3 model outputs used in this study are available: http://gws-access.jasmin.ac.uk/public/pmip4/HADGEM3_LIG_PI/. The authors declare that all other data are available in the paper and its Supplementary Information.

*Author contributions.* RD conducted all analysis. LCS, DS, and MGV oversaw the direction and formulation of the research. MGV carried out the HadGEM3 simulations and supported the analysis of both simulations. DS guided the interpretation of all simulation results. All

authors have read the manuscript and provided comments. RD wrote the bulk of the manuscript with support from LCS and all other authors.

*Competing interests.* The authors declare no competing interests.

*Acknowledgements.* RD and MVG acknowledge support from NERC research grant NE/P013279/1. LCS acknowledges support through NE/P013279/1, NE/P009271/1, and EU-TiPES. The project has received funding from the European Union's Horizon 2020 research and innovation programme under grant agreement No 820970. DS acknowledges support from the NERC-UKESM program. This work used the

ARCHER UK National Supercomputing Service (http://www.archer.ac.uk) and the JASMIN data analysis platform (http://jasmin.ac.uk/).



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
