# Peer review of "The contribution of melt ponds to enhanced Arctic sea-ice melt during the Last Interglacial"

_The Cryosphere, 2021_

## Referee Comment (RC1)

Review: Diamond et al. The contribution of melt ponds to enhanced Arctic sea-ice melt during the last interglacial.

In this manuscript, Diamond et al conduct analysis of HadGEM3 last interglacial (LIG) and pre-industrial (PI) sea ice conditions, with a specific focus on the importance of melt ponds and the associated albedo-feedbacks for driving the difference in sea ice melt between the two time periods. Their study is important, given the need to understand sea-ice conditions in a previously warmer climate as a tool for predicting future conditions under climate change. Furthermore, their research contributes to the scientific understanding of sea ice simulations in global climate models (GCMs) and providing support for proxy estimates of LIG sea ice conditions. Their motivation for using HadGEM3 is clear and the additional comparison to other models in section 3.3 provides further justification and makes for an interesting discussion. They highlight the relevant research questions and aims at the start of each section, which is useful for a longer manuscript such as this. The structure is useful in guiding the reader through the steps the authors took in determining the importance of melt ponds in their study, but does at times become a little overly long or repetitive. The conclusions are succinct and summarise the manuscript and the main findings well, whilst again highlighting the relevance of the study.

The major concern lies in their reporting of previous studies with some very similar results and lack of 1) citations of such results and 2) discussion of how the two studies agree/disagree. There is some overlap or repetition with the Guarino et al. (2020) publication which is concerning and needs addressing. However, Diamond et al do provide a more thorough investigation into the melt ponds and their characteristics. They just need to make their novel results clearer and state explicitly where their results are not novel. There are also a number of minor concerns and inconsistencies throughout the manuscript, and it would benefit from re-wording and cutting down a number of sentences to make their point clearer. A more thorough explanation of a number of methods would also be useful to aid understanding of figures and results. Finally, a number of changes are required to the figures to ensure consistency. Therefore, I recommend major revisions before this article is published.

Major:
Whilst the Guarino et al (2020) study largely compares HadGEM3 with HadCM3, there are still some results concerning the PI and LIG sea ice differences and melt ponds. Some of your results are identical to those in Guarino et al. (2020), which reduces the novelty/originality aspect of the study. You need to focus on novel aspects of the study, of which there are plenty, and create a discussion section to compare your results with other studies. Where your results have been explicitly presented by Guarino et al., you should remove them, or at the least state that these results are from a different study and cite it. Results which were already presented by Guarino et al. (2020) require a discussion/comparison or citation to show that this isn't the first reporting. Specifical examples include:
Guarino: 'In the LIG simulation, the melt season starts earlier, with a max melt pond fraction reaching in mid-June (not shown)'.
Diamond: 'Maximum pond area at 2.67 million km2 is reached in mid-June'. Line 221. Peak sea ice fraction in LIG in early July, compared to peak in late July for PI. Line 225.

Diamond's discussion of spatial differences and sea ice onset/rate is more detailed than Guarino et al. (2020) so there are novel aspects, but some citing of the previous study is missing.

Guarino: 'This confirms that local thermodynamic processes are responsible for the difference between the two models and that melt pond formation plays a key role in determining how much of the additional TOA SW radiation during the LIG can be absorbed by the surface'. 'LIG TOA rad flux is 60-75Wm2 higher than during the PI in early summer. The crucial aspect is to what extent this increase causes additional melt of sea ice. The substantial increase of surface net short-wave flux (with max value of 70Wm-2 in July) is caused by a decrease of surface albedo.'

Diamond: 'we compare incoming SW radiation between LIG and PI, and use this to quantify to what extent ice-albedo feedbacks modify the surface net SW anomaly shown'. Line 246. 'We have demonstrated that the surface net SW anomaly in Fig. 6(a) was caused by the TOA SW anomaly, significantly amplified by stronger LIG than PI ice-albedo feedbacks' Line 252. 'This demonstrates the significant impact of melt ponds on the surface energy balance, and by extension their key role in enhancing LIG summer sea-ice loss.' Line 273.

Again, Diamond does include much more detailed analysis to support their conclusions, however there is no highlighting that some of these results aren't novel.

Guarino reports the total absence of sea ice in September and discusses the importance of this timing, similar to the results in Diamond.

Guarino: 'clouds over sea ice play little role in determining LIG-PI anomalies in SEB of Arctic.'

Diamond: 'Longwave forcings and feedbacks related to cloud cover are not a dominant contributor to the enhanced LIG sea-ice loss.' Line 173.

Guarino: 'The bias towards thick sea ice does not provide protection…LIG summer sea-ice free state in HadGEM3 takes around 5 model-years to complete.'

Diamond: 'An ice-free summer state is reached after only 4 years… preconditioning does not play the dominant role in enhanced melting of sea-ice during the LIG'. Line 157-159.

Fig 1 in Guarino is similar to Fig 2 in Diamond. March, July and September SIC are plotted for both PI and LIG in both publications. Changing the colour bar alone and including other months is not enough. Perhaps you could include an anomaly plot, so that readers can better determine the similarities and differences between the two periods in all months and not just when there is a large difference.

The Guarino et al. (2020) study is not mentioned at all in the results (which also serves as a discussion, as no separate discussion is provided), except for Figure 6 caption.

I suggest that you include a discussion. Section 3.3 could also be included in the discussion.

Minor:

Throughout

Check your tenses throughout the manuscript. Typically, when talking about previous studies or work of others, past tense is used. Some sentences have a combination of past and present. Similarly, be careful of words like 'remarkable', 'simple' etc. These are too subjective.

Introduction

You need to include some connection between the motivation for using HadGEM3 and the melt pond section. Until line 54, you didn't mention the importance of melt ponds or whether anyone else has researched their impact on sea ice loss in GCMs, despite this being

the most crucial aspect of your motivation. I would link melt ponds into the section where you state the mechanisms driving Arctic sea ice change.

Ln 55: Melt ponds 'seem to be critical' for sea ice loss simulations. What evidence do you have of this? As this is partly the motivation for your study, it needs to be clearer. Why did you decide to investigate the impacts of melt ponds?

Ln 68-74: This paragraph is simply a list of previous findings, but with no context or link to your study. I suggest you remove the first few sentences and focus on which melt pond models are included in GCMs, specifically HadGEM3, and simply say that the development of such models comes from a series of studies and improvements. Then simply cite the relevant studies.

Methods:

How is sea ice treated or represented in the PI and LIG simulations? You point the reader to the Ridley et al. paper, but as sea ice is a crucial aspect of your publication, a short paragraph describing some of the key characteristics is required. For instance, melt pond information from which thickness category of sea ice is used? Melt pond fraction and depth are calculated separately for each ice thickness category (Ridley et al. 2018b)

You use the low resolution HadGEM3. How does the model resolution potentially influence your results, especially when looking into melt ponds which are considerably smaller?

I suggest that you move more of your description throughout the results into the methods. For example, include definitions of 'ice covered' and 'open water' grid cells, as these are used throughout.

Results:

Ln 132: Could you show the sea ice thickness, perhaps as a panel next to sea ice area. You mention sea ice thickness at least 4 or 5 times throughout the manuscript, so it would aid the readers understanding.

Ln 142-144: Sentence not clear. Specifically, which season you are talking about.

Ln 145: Your motivation is wanting to know what causes the spring melting. You mention ocean, clouds, albedo, wind, ice distribution- but don't mention melt ponds, despite the introduction and title. Similarly, you don't do any analysis related to ocean heat transport, but only discuss previous studies (line 158-161), so including ocean heat transport in line 145 onwards is misleading. As ocean heat transport is not important nor a result that you found, I would move this to either the introduction or discussion.

Ln 153 onwards: You state that you don't use the spin up years in your methods (Ln 109), but here you look at them. You need to make it clearer in your methods that you don't discard them entirely.

Ln 157: How does the August decline show that preconditioning isn't playing a role? Is this winter preconditioning that you are talking about, as in line 148? In which case, should you not focus on the winter conditions during spin up?

Ln 176: change to 'This suggests..'. You are concluding quite strongly that thermodynamic processes are predominantly from SW anomalies, but you are looking at spin-up and only 1 year. Which I wouldn't say is totally conclusive.

Ln 187: Are ice volume tendencies split by dynamics and thermodynamics directly output by HadGEM3? Or have you calculated these and how? This should be made clear here, and if necessary, included in the methods.

Ln 213: what do you mean by distant from sea ice edge? Aren't areas south of 70°N closer to the sea ice edge?

Ln 219: what do you mean by stronger? Is the rate of change faster? Stronger isn't the right word here.

Ln 236: Where did you get these albedo values from? In the introduction you provide a range for melt ponds from 0.1-0.5, so why use 0.27?

Ln 253: You don't discuss the ice-albedo feedbacks in the above paragraph. You talk about the anomalies in downwelling and total absorbed SW, but you don't separate the total absorbed SW into ice-albedo feedbacks. Are you including absorption by ocean and melt ponds into this statement?

Ln255 section: Are there other things that might have changed the albedo- such as snowfall changes? I am not suggesting that you look into them, but is it possible that albedo changes on the sea ice are not only related to melt ponds?

Ln 288: explain what is meant by thin-ice fraction here.

Ln 285 to 292: this section should be made shorter and clearer, there's a lot of repetition.

Ln 294-300: lots of new acronyms. Are they all necessary? If they're only used in this section, just write them out.

Ln 295-300: There is no discussion of the April corelations with thin-ice being statistically relevant. I would also mention the importance of this variable for prediction, along with where you state the difference in April PI and LIG. I think you try to explain this in line 298-300, but it is not clear, as despite the difference in SIA and thickness between the two time periods, the April correlation is similar in both periods. Perhaps a summary sentence is needed here too- do you see the predictability of SIA from melt ponds changing as the climate warms?

Ln 311: SIE not explained previously, and only used in this section- write out the abbreviation.

Ln 313: What are the other models? Or link to a paper which explains the CMIP6-PMIP4 experiment so readers can see all model options.

Ln 325: melt pond schemes should 'matter' all of the time if we are attempting to simulate the most accurate conditions. I would rephrase this sentence.

Ln 336 (and in other locations): if the grid cell is classed as 'ice covered', how is open water involved? How do you calculate open water? Just where sea ice is less than 0.15? Why is there a difference in open water absorption between the two periods? If the same sea ice area is used? If necessary, I would include some explanation in the methods.

Ln 348: I would say May-June melt pond area is a good predictor, as April isn't.

Ln 350: watch your tenses here and throughout the manuscript. LIG is in the past, as is the simulation, and condense. 'which has ice that is thinner' could be 'which had thinner ice and was thus…'

Figures:
There seems to be inconsistency in your evaluations. Figure 10 is first 50 years of simulations- but in other areas you use 200 years. Why? Figure 12 is 200-year average.
Figure 13: 50 years again.
Figure 5: The legend covers too much of the results from PI.
Figure 6: no a and b labels. Caption is not clear. Reword to: Anomalies (LIG-PI) of the components of the surface energy budget from a) the LIG simulation (adapted..) and b) first year of LIG spin-up. What does 'anomalies are found from long-term means' mean? If figure b is the first year of spin up, what long-term mean can be used? Again, the label is covering your results.

Figure 10: What is your scale in Figure 10? Day of the year? So 90 refers to approx. start of April? I would include some more information in the label or caption.

Figure 12: change to c) shows melt pond fraction of sea ice.

Figure 13 c: What is meant by 'up to this day'? The caption and label are not clear here. Alter y axis labels on a and b to put units in brackets, to be consistent with figure 13c. Figure c needs more ticks and labels for x axis, as you talk about specific dates in the text, which are currently hard to see.

Figure 14: Does ASO mean august, September, October? Only the red square is April, May, June, and the other symbols are August, September, October? Or I the 'for comparison' sentence only for reference and not actually plotted? In that case, remove it from the caption, or add panels to show this. Similarly, as there is plenty of room, label your x axis with the months rather than letters. This would be consistent with figure 13 and others. Full stop missing from last sentence. Explain the three dotted lines in the caption.

Technical:

Ln 27+30: 'find' should be 'found'

Ln 29: 'explore' should be 'explored'

Ln 32-33: reword sentence 'This makes it difficult to determine the mechanisms or distribution of sea ice loss during the LIG from these preserved biological data.'

Ln 46: remove 'however', it is not needed with a sentence starting with 'whilst'

Ln 50: remove 'apparently'

Ln 59: remove extra parenthesis

Ln 96: Above you say CICE5.1 by Hunke et al. 2015 is used, but here it is just called CICE. Is there a difference? If not, include 'hereafter called CICE' in line 87.

Ln 109: Documentation from UKESM states 615 years of spin up, not 700 for HadGEM3 GC3.1 N96ORCA1 (https://ukesm.ac.uk/cmip6-simulations-hadgem3-gc3-1/)

Ln 134: include 'for temperature and salinity respectively' after 1-2ppt.

Ln 134: SST not yet explained.

Ln 141: citation for this?

Ln 141: SSS not explained. This is the only time you use this abbreviation, so I would just use the full words, as you have many abbreviations already in use.

Ln 141: Combine the two sentences: 'Thus, it is additionally useful to consider mean sea surface salinity changes (Fig. 4), which have a clearer signature than SST.'

Ln 145: reword 'we of course wish to know' to 'we aim to understand'

Ln 151: reword to 'The process which is most significant for the simulated enhanced LIG sea ice loss, is currently unknown'.

Ln 169: bracket around a not needed in Figure citation.

Ln 174: include 'likely' or 'possibly' in this sentence, as you have only shown they are not important in the spin-up, not for all sea-ice processes.

Ln 208: 'that' should be 'than'

Ln 209: reword for clarity. Only after a number of reads through this section did I understand this point. Perhaps to: 'As melt onset is not provided as a variable by the models, we use the mean first day each grid-cell has pond fraction greater than 1% as a proxy for this'.

Ln 217: remove simply.

Ln 220: Make clear that this sentence refers to LIG conditions

Ln 247: 'Outlined in 1'- what does that mean? What is 1?

Ln 268: you shouldn't be computing something from a figure… perhaps highlighted or presented?
Ln 270: missing full stop after (c).
Ln 328/341: Lead should be led.
Ln 332: re-amplifies should be amplifies

---

## Author Comment (AC1)

**Response to referee comments**

**Anonymous Referee #1**

We thank Referee #1 for the time they spent reviewing our manuscript, and the helpful comments and feedback they provided. See below our point-by-point response to their comments, with their comments marked in black and our responses marked in blue.

**General comments**

In this manuscript, Diamond et al conduct analysis of HadGEM3 last interglacial (LIG) and pre-industrial (PI) sea ice conditions, with a specific focus on the importance of melt ponds and the associated albedo-feedbacks for driving the difference in sea ice melt between the two time periods. Their study is important, given the need to understand sea-ice conditions in a previously warmer climate as a tool for predicting future conditions under climate change. Furthermore, their research contributes to the scientific understanding of sea ice simulations in global climate models (GCMs) and providing support for proxy estimates of LIG sea ice conditions. Their motivation for using HadGEM3 is clear and the additional comparison to other models in section 3.3 provides further justification and makes for an interesting discussion. They highlight the relevant research questions and aims at the start of each section, which is useful for a longer manuscript such as this. The structure is useful in guiding the reader through the steps the authors took in determining the importance of melt ponds in their study, but does at times become a little overly long or repetitive. The conclusions are succinct and summarise the manuscript and the main findings well, whilst again highlighting the relevance of the study.

The major concern lies in their reporting of previous studies with some very similar results and lack of 1) citations of such results and 2) discussion of how the two studies agree/disagree. There is some overlap or repetition with the Guarino et al. (2020) publication which is concerning and needs addressing. However, Diamond et al do provide a more thorough investigation into the melt ponds and their characteristics. They just need to make their novel results clearer and state explicitly where their results are not novel. There are also a number of minor concerns and inconsistencies throughout the manuscript, and it would benefit from re-wording and cutting down a number of sentences to make their point clearer. A more thorough explanation of a number of methods would also be useful to aid understanding of Figures and results. Finally, a number of changes are required to the Figures to ensure consistency. Therefore, I recommend major revisions before this article is published.

Following these comments, we have made substantial changes to the structure of the manuscript. Primarily, we have restructured the Introduction, with melt pond processes introduced and discussed earlier in the text, to make the motivation for our study clearer. In the Methods section, we have provided more information about the sea ice model. We have also added a more thorough explanation of techniques applied in the Methods section, to aid understanding of the results and figures presented. We have added a Discussion section to compare our results with those of prior publications, in particular the Guarino et al. (2020) paper; this Discussion section also includes the original Section 3.3. Throughout, we have re-worded and cut down text where necessary to make our points clearer. We also plan to improve certain figures, to ensure consistency. Please find details in the point-by-point response below.

**Major:**

Whilst the Guarino et al (2020) study largely compares HadGEM3 with HadCM3, there are still some results concerning the PI and LIG sea ice differences and melt ponds. Some of your results are identical to those in Guarino et al. (2020), which reduces the novelty/originality aspect of the study. You need to focus on novel aspects of the study, of which there are plenty, and create a discussion section to

compare your results with other studies. Where your results have been explicitly presented by Guarino et al., you should remove them, or at the least state that these results are from a different study and cite it. Results which were already presented by Guarino et al. (2020) require a discussion/comparison or citation to show that this isn't the first reporting.

We would like to thank the reviewer for pointing this out. Our intention has been to make the paper readable without knowing the Guarino et al. (2020), following a logical flow by content. However, we agree that it is crucial to clearly state what has been reported before and what is new. Therefore, we have restructured our manuscript, adding a new Discussion section, and changes in the Introduction.

Specifica examples include:

Guarino: 'In the LIG simulation, the melt season starts earlier, with a max melt pond fraction reaching in mid-June (not shown)'.

Diamond: 'Maximum pond area at 2.67 million km2 is reached in mid-June'. Line 221. Peak sea ice fraction in LIG in early July, compared to peak in late July for PI. Line 225.

This sentence was reworded and a citation added as follows.

'Maximum pond area was reached in mid-June (Guarino et al., 2020b), which we compute as 2.67+/-0.37 million km$^2$'

Diamond's discussion of spatial differences and sea ice onset/rate is more detailed than Guarino et al. (2020) so there are novel aspects, but some citing of the previous study is missing.

Guarino: 'This confirms that local thermodynamic processes are responsible for the difference between the two models and that melt pond formation plays a key role in determining how much of the additional TOA SW radiation during the LIG can be absorbed by the surface'. 'LIG TOA rad flux is 60-75Wm2 higher than during the PI in early summer. The crucial aspect is to what extent this increase causes additional melt of sea ice. The substantial increase of surface net short-wave flux (with max value of 70Wm-2 in July) is caused by a decrease of surface albedo.'

Diamond: 'we compare incoming SW radiation between LIG and PI, and use this to quantify to what extent ice-albedo feedbacks modify the surface net SW anomaly shown'. Line 246.

'We have demonstrated that the surface net SW anomaly in Fig. 6(a) was caused by the TOA SW anomaly, significantly amplified by stronger LIG than PI ice-albedo feedbacks' Line 252.

'This demonstrates the significant impact of melt ponds on the surface energy balance, and by extension their key role in enhancing LIG summer sea-ice loss.' Line 273.

Again, Diamond does include much more detailed analysis to support their conclusions, however there is no highlighting that some of these results aren't novel.

We have added citations of Guarino et al. (2020) where relevant in Results, and have added a comparison between our results and that of Guarino et al. in Discussion.

In Results:

'As the LIG-PI albedo difference increased so significantly from April through June, we have shown that the surface net SW anomaly in Fig. 6a was caused by the TOA SW anomaly, very likely significantly amplified by stronger LIG than PI albedo feedbacks, as first suggested in Guarino et al. (2020b).'

In Discussion:

'Guarino et al. (2020b) found that HadGEM3 simulated an ice-free summer LIG Arctic, and suggested this was linked to HadGEM3's realistic representation of melt pond physics. Here, we have quantified the importance of melt ponds during the LIG in depth, showing that greater melt pond formation, earlier in the year, directly led to the large surface energy budget difference between LIG and PI that was first highlighted in Guarino et al. (2020b). We have demonstrated the impact of melt ponds on thermodynamic processes during the LIG, and their role in determining LIG spring albedo over ice-covered regions, leading to albedo feedbacks and ultimately the extreme ice loss first presented in Guarino et al. (2020b). Additionally, our analysis of the HadGEM3 PMIP4 LIG spin-up simulation years, showing that changes to longwave radiation and ocean heat transport were not primary drivers of the

observed enhanced LIG ice loss, was supported by Guarino et al. (2020b), who used the LIG production run to show negligible differences in these factors between the simulated LIG and PI.'

Guarino reports the total absence of sea ice in September and discusses the importance of this timing, similar to the results in Diamond.

Guarino: 'clouds over sea ice play little role in determining LIG-PI anomalies in SEB of Arctic.'

Diamond: 'Longwave forcings and feedbacks related to cloud cover are not a dominant contributor to the enhanced LIG sea-ice loss.' Line 173.

We have added citations of Guarino et al. (2020) where relevant in Results, and have compared our results and that of Guarino et al. in Discussion (see above).

In Results:

'We note also that the longwave anomaly accounted for < 5 Wm$^2$ of the total surface heat flux anomaly in Fig. 6b, so longwave forcings and feedbacks related to cloud cover were likely not dominant contributors to the enhanced LIG sea-ice loss, which agrees with the findings of Guarino et al. (2020b).'

In Discussion:

See above.

Guarino: 'The bias towards thick sea ice does not provide protection…LIG summer sea-ice free state in HadGEM3 takes around 5 model-years to complete.'

Diamond: 'An ice-free summer state is reached after only 4 years… preconditioning does not play the dominant role in enhanced melting of sea-ice during the LIG'. Line 157-159.

We have added a citation of Guarino's finding that a sea-ice free state is reached within around 5 years in Results, as follows:

'However, an ice-free summer state was reached after only 4 years (as found by Guarino et al., 2020b): we find August sea-ice area halved during the first year of the spin-up run from nearly 6 million to around 3 million km$^2$.'

Fig 1 in Guarino is similar to Fig 2 in Diamond. March, July and September SIC are plotted for both PI and LIG in both publications. Changing the colour bar alone and including other months is not enough. Perhaps you could include an anomaly plot, so that readers can better determine the similarities and differences between the two periods in all months and not just when there is a large difference.

Fig. 2 is important to understand our manuscript, but we indicate in revised figure what has been shown previously. In addition, we will add a figure showing the LIG-PI anomalies of March-September SIC in the Supplementary Information.

The Guarino et al. (2020) study is not mentioned at all in the results (which also serves as a discussion, as no separate discussion is provided), except for Fig. 6 caption.

I suggest that you include a discussion. Section 3.3 could also be included in the discussion.

Good advice. We have restructured and added additional detail to the Introduction, to introduce the Guarino et al. (2020) study earlier on for context. We have added a Discussion section, to contrast our results with those of prior publications (in particular, those of Guarino et al. (2020) – see above). The Discussion section also includes the original Section 3.3.

**Minor:**
**Throughout**

Check your tenses throughout the manuscript. Typically, when talking about previous studies or work of others, past tense is used. Some sentences have a combination of past and present. Similarly, be careful of words like 'remarkable', 'simple' etc. These are too subjective.

Thank you. We have corrected tenses throughout the manuscript, and changed words that were too subjective.

**Introduction**

You need to include some connection between the motivation for using HadGEM3 and the melt pond section. Until line 54, you didn't mention the importance of melt ponds or whether anyone else has researched their impact on sea ice loss in GCMs, despite this being the most crucial aspect of your motivation. I would link melt ponds into the section where you state the mechanisms driving Arctic sea ice change.

We have substantially restructured the Introduction as suggested, with ponds introduced and discussed earlier in the text, now at line 30 instead of line 54. We have added to the Introduction additional context describing HadGEM3's representation of melt ponds.

Ln 55: Melt ponds 'seem to be critical' for sea ice loss simulations. What evidence do you have of this? As this is partly the motivation for your study, it needs to be clearer. Why did you decide to investigate the impacts of melt ponds?

We have addressed this point by providing additional context discussing melt ponds and their representation in sea ice loss simulations in the Introduction, and linking the motivation for our study to the Guarino et al. (2020) paper.

Ln 68-74: This paragraph is simply a list of previous findings, but with no context or link to your study. I suggest you remove the first few sentences and focus on which melt pond models are included in GCMs, specifically HadGEM3, and simply say that the development of such models comes from a series of studies and improvements. Then simply cite the relevant studies.

We have changed this paragraph as suggested.

In Introduction:

'In spite of their importance, melt ponds have only rather recently been explicitly included in CMIP models. In most CMIP6 models, the most common approach is to implicitly parameterise melt ponds by reducing the ice/snow albedo when surface ice temperatures approach 0 °C (e.g. Collins et al., 2006; Curry et al., 2001). This tuning has been relatively successful for reproducing realistic melt rates for the present day (Collins et al., 2006; Curry et al., 2001). However, pond formation is affected by sea-ice processes throughout melt season (e.g. evolving topography and snow cover), which this tuning does not represent (Kwok et al., 2009). Therefore, in recent years, there has been increasing interest in incorporating more detailed melt pond models into GCMs. HadGEM3 includes one of the most comprehensive, to date, melt pond schemes in its sea-ice component CICE5.1 ((Hunke et al., 2015), detailed in Section 2), a result of a series of developments and improvements (Taylor and Feltham, 2004; Lüthje et al., 2006; Skyllingstad et al., 2009; Scott and Feltham, 2010; Flocco et al., 2012).'

**Methods:**

How is sea ice treated or represented in the PI and LIG simulations? You point the reader to the Ridley et al. paper, but as sea ice is a crucial aspect of your publication, a short paragraph describing some of the key characteristics is required. For instance, melt pond information from which thickness category of sea ice is used? Melt pond fraction and depth are calculated separately for each ice thickness category (Ridley et al. 2018b)

We have added the necessary additional detail to the section Methods: Model, as follows:

'The modifications and setup of the applied sea-ice model CICE 5.1 (hereafter, 'CICE') are described in Ridley et al. (2018b). The standard elastic–viscous–plastic rheology (EVP) has been applied for ice dynamics with default CICE remapping advection algorithm and ridging schemes (Hunke et al., 2015). Ice thermodynamics are based on Bitz and Lipscomb (1999) with 4 ice and 1 snow layer. A semi-implicit coupling scheme between atmosphere and sea-ice has been introduced to ensure the stability of the solver West et al. (2016). The evolution of the sea-ice is separately calculated for 5 ice thickness categories within each grid cell.'

You use the low resolution HadGEM3. How does the model resolution potentially influence your results, especially when looking into melt ponds which are considerably smaller?

The model does not simulate individual melt ponds, but the area which is covered by melt ponds accounting for ice topography. Thus, the model resolution has no impact regarding melt ponds. No medium and high resolution runs for the Last Interglacial period exist. While there are some differences between the historical HadGEM3 simulations with medium and low resolution, results are not expected to change significantly.

I suggest that you move more of your description throughout the results into the methods. For example, include definitions of 'ice covered' and 'open water' grid cells, as these are used throughout.

We have added an additional 'Analysis' subsection to Methods, to define key terms and describe analysis techniques used, and have moved most of the description of techniques in Results to this section.

In Analysis:

'For a given ice-covered cell (defined as a cell with sea-ice concentration> 0.15), the area of the cell at a subgrid scale of (a) bare ice and snow (b) melt ponds on ice and (c) ocean that is exposed to the atmosphere and not covered by ice (hereafter, the 'open water' area), may be computed as follows. The area of the cell covered by ice and snow (including pond-covered ice), and (b), are returned as model output variables. (c) is then computed as the grid cell area not covered by ice and snow, and (a) is the ice area not covered by ponds.

Unless otherwise indicated in the figure caption, figures showing the daily climatology of a variable are calculated using only ice-covered grid-cells above 70 °N, and the 200-year mean of the data is shown with the error as the standard deviation (reflecting the interannual variability) for each day of the year.'

**Results:**

Ln 132: Could you show the sea ice thickness, perhaps as a panel next to sea ice area. You mention sea ice thickness at least 4 or 5 times throughout the manuscript, so it would aid the readers understanding.

A figure showing the LIG and PI sea ice thickness from March-September will be added below the sea ice area figure.

Ln 142-144: Sentence not clear. Specifically, which season you are talking about.

This sentence has been modified as follows: "These have a clearer signature than SST. While the salinity patterns were similar in May for both simulations, the stronger melt during the LIG caused the region with LIG winter ice cover to become significantly fresher than the PI, by around 1-2ppt in June and July …"

Ln 145: Your motivation is wanting to know what causes the spring melting. You mention ocean, clouds, albedo, wind, ice distribution- but don't mention melt ponds, despite the introduction and title.

We have added the sentence: 'Ice-albedo feedbacks may be significantly impacted by the presence of melt ponds, as discussed in Section 1'.

Similarly, you don't do any analysis related to ocean heat transport, but only discuss previous studies (line 158-161), so including ocean heat transport in line 145 onwards is misleading. As ocean heat transport is not important nor a result that you found, I would move this to either the introduction or discussion.

Fig 5 strongly implies that change to ocean heat transport is not an important driver of the LIG sea-ice loss, as summer sea-ice is lost so rapidly during the spin-up, so some discussion of ocean heat transport should be included here.

We have rephrased the paragraph to make this clearer, as follows:

'Similarly, this rapid loss of sea-ice within the first few spin-up years cannot be explained by change to ocean heat transport, which is often linked to reduced sea-ice (Holland et al., 2006; Steele et al., 2010): upper ocean heat transport takes decades to equilibrate in numerical models after a significant perturbation, and deeper ocean heat transport, centuries to millenia (Kantha and Clayson, 2000). Therefore, from Fig 5, changes to ocean heat transport could not have been the first-order driver of the rapid LIG sea-ice loss observed.'

Our results are supported by conclusions in Guarino (2020) that ocean heat transport is unaltered between the LIG and PI simulations, which we have now mentioned in Discussion (see above)

Ln 153 onwards: You state that you don't use the spin up years in your methods (Ln 109), but here you look at them. You need to make it clearer in your methods that you don't discard them entirely.

We have added for clarity in Methods: 'The first 35 years of LIG spin-up, and all 200 years of the LIG production run, are used in our analysis.'

Ln 157: How does the August decline show that preconditioning isn't playing a role? Is this winter preconditioning that you are talking about, as in line 148? In which case, should you not focus on the winter conditions during spin up?

We will rephrase the paragraph. Our point is that the differences between PI and LIG do not develop slowly over time (by e.g. reducing ice thickness a bit every year), but rapidly during first summer.

Ln 176: change to 'This suggests..'. You are concluding quite strongly that thermodynamic processes are predominantly from SW anomalies, but you are looking at spin-up and only 1 year. Which I wouldn't say is totally conclusive.

Thank you. We have changed as suggested.

Ln 187: Are ice volume tendencies split by dynamics and thermodynamics directly output by HadGEM3? Or have you calculated these and how? This should be made clear here, and if necessary, included in the methods.

We have added for clarification that both ice volume tendencies split by dynamics and thermodynamics are 'primary model output variables'

Ln 213: what do you mean by distant from sea ice edge? Aren't areas south of 70°N closer to the sea ice edge?

We have removed 'distant from sea ice edge' for clarity

Ln 219: what do you mean by stronger? Is the rate of change faster? Stronger isn't the right word here.

We have changed this sentence to 'At the end of April, the rate of pond formation was greater during the LIG than the PI.'

Ln 236: Where did you get these albedo values from? In the introduction you provide a range for melt ponds from 0.1-0.5, so why use 0.27?

0.27 is the value the model uses. We have added the relevant albedo values in the Methods section, as well as the relevant citation.

In Methods:

'For our study it is important to mention that the albedo calculation is based on the scheme used in the CCSM3 model (Hunke et al., 2015), but includes surface melt ponds by applying the explicit topographic melt pond model of Flocco et al. (2012) and Flocco et al. (2010). Meltwater, formed as a result of snow melt, ice melt and precipitation, runs downhill under the influence of gravity and collects on sea-ice starting at the lowest surface height applying the sub-grid scale ice thickness distribution. The evolution of pond fraction and depth as well as the formation of ice lids are calculated. The albedo of ponds of depth>20cm is 0.27 (Ridley et al., 2018b), significantly smaller than the albedo of ice and snow, which varies through the year in the range 0.5-0.9 and is calculated as described in Hunke et al. (2015). The albedo of open water is 0.07+/-0.03 at the latitudes and over the months for which the open water albedo is considered in our analysis (Ridley et al., 2018b). '

Ln 253: You don't discuss the ice-albedo feedbacks in the above paragraph. You talk about the anomalies in downwelling and total absorbed SW, but you don't separate the total absorbed SW into ice-albedo feedbacks.

To clarify the point about ice-albedo feedbacks: we have modified the last sentence of the previous section (line 244-245) to read 'In May, the gradient of the surface downwelling SW flux decreased, but the gradient of the surface net SW flux increased. This increasing rate of ice melt (see also Fig 1), despite

the decreasing rate of change of incoming solar radiation, implies that albedo feedbacks played a strong role in SW absorption through May.'

We have also modified the first sentence of this section (line 246-247): 'Using Fig 13b, we compare incoming SW radiation between LIG and PI, and use this to quantify to what extent albedo changes (related to albedo feedbacks) modified the surface net SW anomaly shown in Fig. 6.'

We have also modified the final sentence of the section (line 252-253): 'We have demonstrated that the surface net SW anomaly in Fig. 6a was caused by the TOA SW anomaly, very likely significantly amplified by stronger LIG than PI albedo feedbacks, as the LIG-PI albedo difference increased so significantly from April through July.'

Are you including absorption by ocean and melt ponds into this statement?

Yes: at first we consider the total change in absorption, and then break this down into absorption due to each component in following sections to show that melt ponds have significant impact. To clarify this, we have modified the sentence (line 238) to: 'The change in albedo (due to changes in coverage of bare ice, snow, ponds and open water) can be compared between the LIG and PI.'

Ln255 section: Are there other things that might have changed the albedo- such as snowfall changes? I am not suggesting that you look into them, but is it possible that albedo changes on the sea ice are not only related to melt ponds?

Fig. 13b provides a detailed analysis of the changes in absorbed SW radiation between LIG and PI: anomalies for melt ponds, open water and ice and snow cover (including changes due to snowfall). E.g. for beginning of June, melt ponds absorb 35 Wm$^{-2}$ additional radiation compared to 8 Wm$^{-2}$ for ice and snow cover and 20 Wm$^{-2}$ for the increased lead fraction.

Ln 288: explain what is meant by thin-ice fraction here.

We have added an explanatory note: thin ice is 'defined as fraction of ice-covered grid-cells with ice thickness<1.4m'

Ln 285 to 292: this section should be made shorter and clearer, there's a lot of repetition.

We have shortened and clarified this section as follows:

'For the 200 years' of simulation output for both LIG and PI, Pearson's correlation coefficient was calculated between the mean August to October sea-ice area, and each of the following four variables in each of April, May and June:
(1) the mean monthly melt pond fraction
(2) the 'radiation-effective pond fraction', denoting the fraction of grid cell area covered by ponds that are not covered by an ice lid and thus are expected to affect the surface albedo (DuVivier, 2018)
(3) the thin-ice fraction, defined as fraction of sea-ice of thickness 1.4 m, as the ice state in spring is known to affect the summer sea-ice area (Schröder et al., 2014)
(4) the fraction of incoming short-wave radiation absorbed, as this fraction over ice-covered cells accounts for albedo changes from open water as well as ponds in these cells.'

Ln 294-300: lots of new acronyms. Are they all necessary? If they're only used in this section, just write them out.

We have written out the acronyms.

Ln 295-300: There is no discussion of the April corelations with thin-ice being statistically relevant. I would also mention the importance of this variable for prediction, along with where you state the difference in April PI and LIG. I think you try to explain this in line 298-300, but it is not clear, as despite the difference in SIA and thickness between the two time periods, the April correlation is similar in both periods.

We have added discussion of thin-ice fraction as a predictor for both time periods: 'We note that, for both time periods, the April through June thin-ice fraction is a statistically significant (p<10$^{-5}$) predictor of summer ice area, as might be expected (Schröder et al., 2014). However, of more interest here is the significance of the pond-related correlations through the spring'.

Perhaps a summary sentence is needed here too- do you see the predictability of SIA from melt ponds changing as the climate warms?

We have added a summary sentence: 'This is of importance for future seasonal predictions: as the climate warms and Arctic sea-ice continues to thin, the spring melt pond area is likely to become an increasingly reliable predictor of September sea-ice area.'

Ln 311: SIE not explained previously, and only used in this section- write out the abbreviation.

We have written out the abbreviation.

Ln 313: What are the other models? Or link to a paper which explains the CMIP6-PMIP4 experiment so readers can see all model options.

We have added relevant citations to this paragraph.

Ln 325: melt pond schemes should 'matter' all of the time if we are attempting to simulate the most accurate conditions. I would rephrase this sentence.

We have rephrased this sentence: 'This shows that melt pond formation has a crucial impact on sea-ice in warm climates, potentially making the difference between ice-covered and ice-free summer conditions. Specifically, we have answered the question of how HadGEM3's melt pond scheme contributes to the simulated ice loss in the Arctic during the LIG.'

Ln 336 (and in other locations): if the grid cell is classed as 'ice covered', how is open water involved? How do you calculate open water? Just where sea ice is less than 0.15? Why is there a difference in open water absorption between the two periods? If the same sea ice area is used? If necessary, I would include some explanation in the methods.

We have added further detail in the Methods section to clarify techniques used, with an explanatory section in Methods subsection: Analysis (see above).

Ln 348: I would say May-June melt pond area is a good predictor, as April isn't.

Thank you. Modified.

Ln 350: watch your tenses here and throughout the manuscript. LIG is in the past, as is the simulation, and condense. 'which has ice that is thinner' could be 'which had thinner ice and was thus…'

We have fixed inconsistency in the tenses.

**Figures:**

There seems to be inconsistency in your evaluations. Fig. 10 is first 50 years of simulations- but in other areas you use 200 years. Why? Fig. 12 is 200-year average.

Due to the small interannual variability within each time period, the 50-year means are shown in this Fig. as there is negligible difference from the 200-year mean. For consistency, we will show the 200-year average in this figure.

Fig. 13: 50 years again.

See above. For consistency, we will show the 200-year average in this figure.

Fig. 5: The legend covers too much of the results from PI.

This will be changed.

Fig. 6: no a and b labels. Caption is not clear. Reword to: Anomalies (LIG-PI) of the components of the surface energy budget from a) the LIG simulation (adapted..) and b) first year of LIG spin-up. What does 'anomalies are found from long-term means' mean? If Fig. b is the first year of spin up, what long-term mean can be used? Again, the label is covering your results.

a and b labels will be added label position will be changed so the results can be seen. The caption was reworded as suggested to make it clearer:

'Anomalies (LIG-PI) of the components of the surface energy budget from a) the LIG simulation (adapted from Guarino et al., 2020b) and b) first year of LIG spin-up. For the LIG and PI periods, the 200-year mean was used. For the LIG, PI, and the first year of the LIG spin-up period, the spatial average was computed from monthly data over the region from 70°N - 90°N for the simulated short-wave

radiation, long-wave radiation, sensible heat flux, and latent heat flux. The total surface heat flux anomaly (black) is the sum of these four heat budget anomalies.'

Fig. 10: What is your scale in Fig. 10? Day of the year? So 90 refers to approx. start of April? I would include some more information in the label or caption.

Labels by month will be added to the scale (e.g. day 90 will be labelled 'April 1')

Fig. 12: change to c) shows melt pond fraction of sea ice.

This change was made.

Fig. 13 c: What is meant by 'up to this day'? The caption and label are not clear here. Alter y axis labels on a and b to put units in brackets, to be consistent with Fig. 13c. Fig. c needs more ticks and labels for x axis, as you talk about specific dates in the text, which are currently hard to see.

Fig. 13 c will be moved to supplementary information, and the other suggested changes will be made.

Fig. 14: Does ASO mean august, September, October? Only the red square is April, May, June, and the other symbols are August, September, October? Or is the 'for comparison' sentence only for reference and not actually plotted? In that case, remove it from the caption, or add panels to show this. Similarly, as there is plenty of room, label your x axis with the months rather than letters. This would be consistent with Fig. 13 and others. Full stop missing from last sentence. Explain the three dotted lines in the caption.

The labels will be changed as suggested. We have reworded the caption to make it clearer, and have explained the dotted lines in the caption, as follows:

'Figure 14. Correlation between spring melt pond fraction (MPF) of sea-ice area (SIA) and autumn SIA for (a) PI, and (b) LIG. For both (a) and (b), 200 years of monthly model data were used over the region 70°N - 90°N. Pearson's correlation coefficient R was calculated between mean August-October SIA and, in each of April, May and June of the same year (1) mean pond fraction of sea-ice area (red) (2) the radiation-effective pond-fraction (gold) (3) mean thin-ice fraction of ice area (green; defined as fraction of ice-covered grid-cells with ice thickness<1.4m) and (4) mean fraction of incident short-wave radiation absorbed (grey).

A statistically significant correlation is defined as correlation with $p<0.05$; dotted lines delimit regions of very high statistical significance with $p<0.01$ and $p<0.00001$.'

**Technical:**

Ln 27+30: 'find' should be 'found'

Thank you. Modified.

Ln 29: 'explore' should be 'explored'

Thank you. Modified.

Ln 32-33: reword sentence 'This makes it difficult to determine the mechanisms or distribution of sea ice loss during the LIG from these preserved biological data.'

Thank you. Modified.

Ln 46: remove 'however', it is not needed with a sentence starting with 'whilst'

Thank you. Modified.

Ln 50: remove 'apparently'

Thank you. Modified.

Ln 59: remove extra parenthesis

Thank you. Modified.

Ln 96: Above you say CICE5.1 by Hunke et al. 2015 is used, but here it is just called CICE. Is there a difference? If not, include 'hereafter called CICE' in line 87.

We have added 'hereafter called CICE'.

Ln 109: Documentation from UKESM states 615 years of spin up, not 700 for HadGEM3 GC3.1 N96ORCA1 (https://ukesm.ac.uk/cmip6-simulations-hadgem3-gc3-1/)

Thank you – this was a mistake and we have changed this.

Ln 134: include 'for temperature and salinity respectively' after 1-2ppt.

Thank you. Modified.

Ln 134: SST not yet explained.

We have defined SST.

Ln 141: citation for this?

We have added the citation:

*'Serreze, M. C., and R. G. Barry (2011), Processes and impacts of Arctic amplification: A research synthesis, Global Planet. Change, 77(1), 85–96, doi:10.1016/j.gloplacha.2011.03.004'*

Ln 141: SSS not explained. This is the only time you use this abbreviation, so I would just use the full words, as you have many abbreviations already in use.

We have defined SSS.

Ln 141: Combine the two sentences: 'Thus, it is additionally useful to consider mean sea surface salinity changes (Fig. 4), which have a clearer signature than SST.'

Thank you. Modified.

Ln 145: reword 'we of course wish to know' to 'we aim to understand'

Thank you. Modified.

Ln 151: reword to 'The process which is most significant for the simulated enhanced LIG sea ice loss, is currently unknown'.

Thank you. Modified.

Ln 169: bracket around a not needed in Fig. citation.

We have made this change, and removed the bracket elsewhere in the main body of text for consistency.

Ln 174: include 'likely' or 'possibly' in this sentence, as you have only shown they are not important in the spin-up, not for all sea-ice processes.

Thank you. Modified.

Ln 208: 'that' should be 'than'

Thank you. Modified.

Ln 209: reword for clarity. Only after a number of reads through this section did I understand this point. Perhaps to: 'As melt onset is not provided as a variable by the models, we use the mean first day each grid-cell has pond fraction greater than 1% as a proxy for this'.

Thank you. Modified as suggested.

Ln 217: remove simply.

Thank you. Modified.

Ln 220: Make clear that this sentence refers to LIG conditions

We have added 'LIG'

Ln 247: 'Outlined in 1'- what does that mean? What is 1?

We have changed this to 'Outlined in Section 1'

Ln 268: you shouldn't be computing something from a Fig.... perhaps highlighted or presented?

We have changed 'computed' to 'shown'

Ln 270: missing full stop after (c).

This was added.

Ln 328/341: Lead should be led.

Thank you. Modified.

Ln 332: re-amplifies should be amplifies

Thank you. Modified.

---

## Author Comment (AC2)

Response to referee comments

Anonymous Referee #2

We thank Referee #2 for the time they spent reviewing our manuscript, and the helpful comments and feedback they provided. See below our point-by-point response to their comments, with their comments marked in black and our responses marked in blue.

**Review of Diamond et al.,**

Diamond et al. analyse the processes leading to Arctic sea-ice loss during the Last Interglacial period (LIG) in the HadGEM3 model. The HadGEM3 is the only model simulating an ice-free Arctic in summer at the LIG. The authors analysis suggests that the sea-ice loss is due to thermodynamic processes, and particularly the creation of melt ponds in late spring/early summer. While the HadGEM3 Arctic sea ice LIG results were presented in Guarino et al., 2020, this study looks in more details at the processes at play. As such, this is an interesting study, well suited for TC, however, I find it a bit hard to follow, with many missing links between sentences/paragraphs/sections. I detail a few comments below but encourage the authors to read their manuscripts and improve the flow and links, particularly in the Introduction and the first part of the results.

Following these comments, we have made substantial changes to the structure of the manuscript, which can be seen from the point-by-point response below. Primarily, we have restructured and added context to the Introduction and added a Discussion section, to contrast our results with those of prior publications. We have added further detail in the Methods and Results sections and improved the flow and links.

Section 3.1 (before 3.1.1, as a side note, maybe the structure of the results need to be amended):
- A first short paragraph and Fig. showing a comparison between observed and simulated monthly Arctic sea-ice area would be useful.
  Figures comparing observed and simulated monthly Arctic sea-ice area are shown in Guarino et al. (2020) and Kageyama et al. (2021), which are cited in the Introduction.
- Since the anomalous Arctic sea-ice loss is due to the higher seasonal insolation at the LIG compared to PI, it would be useful to show in Fig. 1 high northern latitude insolation in PI and LIG. The addition of the insolation curve could also lead to a much earlier discussion of the conundrum: i.e. the maximum insolation anomalies are in June while the maximum sea-ice anomalies are reached in August. An earlier statement on that issue could make the flow of the paper easier to understand.
  Figures showing the higher seasonal LIG than PI insolation are shown in Guarino et al. (2020) and Kageyama et al. (2021), which are cited in the Introduction.

Fig. 6 clearly shows the maximum short-wave anomalies in June, yet I find it surprising that during the early years of the spin up (Fig. 5), Aug. and Sept are the ones decreasing sharply and quickly while July (and June) sea-ice decreases less and more slowly.

As for present-day sea-ice, LIG sea-ice had minimum area in August-October. As quasi-equilibrium was reached within the first few years of the LIG spin-up, it should not be surprising that Aug and Sept sea-ice area decreased more than June and July sea-ice area.

As such, I am not so convinced of the usefulness of Fig. 5 (which could be moved to SI?), but wonder if it would make sense to instead show the timeseries of meltpond area in May and June in the spin up (i.e. similar to Fig. 5 but for meltpond area).

This section considers differences between PI and LIG sea-ice and upper ocean. It is important to show the large differences in sea-ice behaviour before introducing melt ponds as a driver of these differences, which we do later in the manuscript.

Another option would be to show sea-ice volume instead of area. Indeed, the July sea-ice area is 50% smaller in LIG than PI, but it is also much thinner, and sparse.

Sea-ice area and volume show similar patterns, but we can make a figure showing sea-ice volume.

It is quite interesting to show the SST and SSS changes in the Arctic, however this is very briefly mentioned and I don't find the flow of the beginning of section 3.1. logical.

It is important to show the simulated SST and SSS changes in the Arctic, as observations of these quantities from proxy records can be used to deduce LIG ice conditions. We include SST and SSS changes in Section 3.1 as they are closely related to the enhanced sea-ice loss that we detail here.

We have modified the introductory paragraph of this section (line 126) to improve the flow:

'Here, we examine the HADGEM3-simulated LIG sea-ice and upper ocean, in order to identify factors that contributed to LIG summer sea-ice loss. We first compare LIG and PI sea-ice area, as well as sea surface temperatures and salinities, which were closely related to sea-ice area in each time period. We then consider the LIG production run and spin-up to determine the primary drivers of LIG sea-ice loss.'

Section 3.1.1: How were the ice volume tendency due to thermodynamics and dynamics calculated (Figs. 7 and 8)?

These were calculated from model output variables, so we have added a sentence describing this.

Section 3.1.2:

There are a lot of Figures (and sub-panel) in the manuscript, and I wonder if all are necessary.

For example, are Figures 10a,b necessary?

Figures 10 a and b are useful to show where melt ponds form in each time period, as readers will not necessarily be familiar with this.

Similarly, wouldn't only fig.12a be sufficient?

Figures 12b and c are useful to compare the initial rates of melt pond formation between LIG and PI, as these are difficult to see in Fig. 12a alone. We can move Figures 12b and c to Supplementary Information.

Maybe only Fig. 13b is necessary.

Fig. 13a is useful to see absolute values of where incoming shortwave radiation is absorbed. Fig. 13c will be moved to the Supplementary Information. Fig. 13d is useful to visualise how Figs 13a and b were computed, but can be moved to the Supplementary Information.

P16, L. 240-255: PI SW as well as albedo differences between LIG and PI are discussed while not shown. Maybe at least the albedo could be shown in Fig. 13 instead.

A subfigure in Fig. 13 showing the albedo climatology for each time period will be added.

**Minor points:**

P1, L. 20: "affect"

Thank you. Modified.

P1, L. 24: what are you referring to here?

We are referring to surface energy balance differences, as mentioned in previous sentence. We have changed the wording to 'surface energy balance differences'

P3, L. 68: Please add a transition sentence before that section

We have restructured the Introduction so this comment no longer applies.

P4, L. 120 and 122: I doubt the model is "in equilibrium" after 350 years of spin-up. Please modify to "quasi-equilibrium" or "the surface variables have reached a steady state".

We have changed the wording to 'quasi-equilibrium'

P6, L. 151: remove "is"

We have reworded this sentence.

P6, L. 153: "year" or "years"?

'Year', as we briefly discuss sea-ice changes over first 35 years of spin-up (Fig 5), but focus in particular on the first year of spin-up (Figs 5 and 6).

P6, L. 154: "Fig. 5 shows that…"

Thank you. Modified.

P16: Many full stops are missing.

We have corrected this.

P18: Section 3.1. discusses "sea-ice area" whereas section 3.3. discusses "sea-ice extent". It would be good to only discuss one or the other.

We have changed this so that Section 3.3 also discusses sea-ice area, rather than sea-ice extent.